# Towards Unsupervised Domain Bridging via Image Degradation in Semantic Segmentation

**Wangkai Li[1], Rui Sun[1], Huayu Mai[1], Tianzhu Zhang[1,2*]**
[1]MoE Key Laboratory of Brain-inspired Intelligent Perception and Cognition,
University of Science and Technology of China
[2]National Key Laboratory of Deep Space Exploration, Deep Space Exploration Laboratory
{lwklwk, issunrui, mai556}@mail.ustc.edu.cn, tzzhang@ustc.edu.cn

## Abstract

Semantic segmentation suffers from significant performance degradation when the trained network is applied to a different domain. To address this issue, unsupervised domain adaptation (UDA) has been extensively studied. Despite the effectiveness of selftraining techniques in UDA, they still overlook the explicit modeling of domain-shared feature extraction. In this paper, we propose DiDA, an unsupervised domain bridging approach for semantic segmentation. DiDA consists of two key modules: (1) Degradation-based Intermediate Domain Construction, which creates continuous intermediate domains through simple image degradation operations to encourage learning domain-invariant features as domain differences gradually diminish; (2) Semantic Shift Compensation, which leverages a diffusion encoder to disentangle and compensate for semantic shift information with degraded time-steps, preserving discriminative representations in the intermediate domains. As a plug-and-play solution, DiDA supports various degradation operations and seamlessly integrates with existing UDA methods. Extensive experiments on multiple domain adaptive semantic segmentation benchmarks demonstrate that DiDA consistently achieves significant performance improvements across all settings. Code is available at https://github.com/Woof6/DiDA.

## 1  Introduction

Semantic segmentation, a fine-grained pixel-wise classification task, assigns semantic class labels to each pixel, facilitating high-level image analysis. Despite the remarkable progress made in this field [56, 6, 18, 17], networks trained within source domain often encounter significant performance degradation when applied to a target dataset due to domain discrepancies. Mitigating this issue to enhance the generalization capability of networks remains a formidable challenge. To address this problem, extensive research has resorted to unsupervised domain adaptation (UDA), which aims to transfer knowledge from a labeled source domain to an unlabeled target domain.

In previous works, to fully exploit the abundance of unlabeled target domain data, self-training techniques have been naturally incorporated into UDA tasks and have emerged as a mainstream paradigm. The core idea of this paradigm lies in constructing a teacher network via temporal ensembling mechanisms, which generates pseudo-labels by predicting the target domain images. These pseudo-labels are then used to progressively guide the student network's learning on the target domain. Despite achieving impressive results, these methods still overlook the explicit modeling of domain-shared feature extraction, which remains a central challenge in UDA. To illustrate this, we refer to the classic notion of causal representation learning [36, 74]: any observed feature can

---

*Corresponding author

39th Conference on Neural Information Processing Systems (NeurIPS 2025).

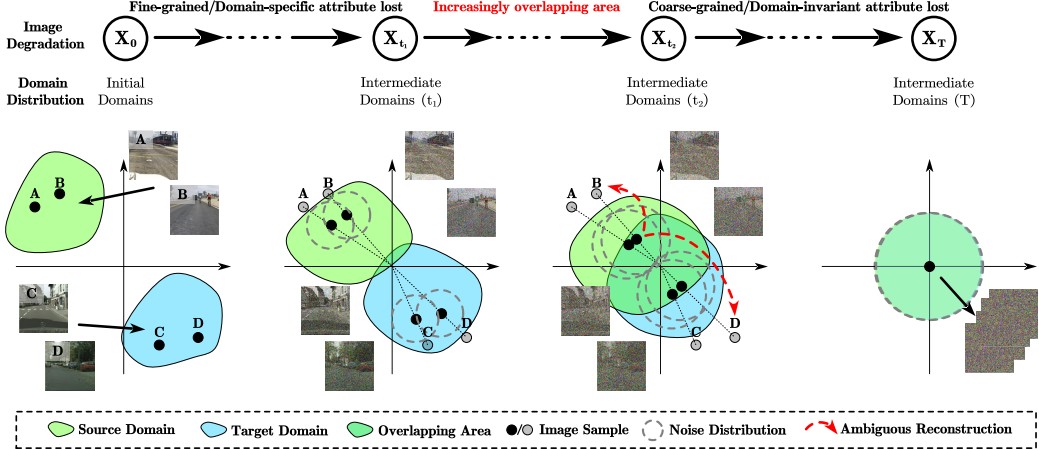

Figure 1: Conceptual illustration of the diffusion forward process. Fine-grained, domain-specific attributes such as texture are lost with less noise added (i.e., early time-steps), while coarse-grained, domain-invariant ones such as shape are lost by adding more noise (i.e., late time-steps).

be generated as $x = \Phi(c, e)$, where $\Phi$ is a generator, $c$ denotes the causal feature that determines the domain-invariant class identity (e.g., shape), and $e$ represents the environment-specific feature (e.g., texture). Since environmental features are domain-specific, the domain shift $e_S \neq e_T$ leads to $x_S \neq x_T$, thereby hindering the learning of a truly domain-invariant class representation.

In this work, we explore a novel perspective for unsupervised domain bridging and demonstrate that simple image degradation operations can serve as effective priors. Our key idea is motivated by the theoretical insight (Sec.3.2) of the Denoising Diffusion Probabilistic Model (DM) [37, 76]. As shown in Fig. 1, the forward diffusion process incrementally adds Gaussian noise to input samples at each time step. This process gradually removes domain-specific attributes, effectively collapsing samples from different domains into a shared representation space. As the time step increases, this diffusion process enlarges the overlapping area of the probability density functions of the noisy domain distributions. Existing studies [95, 77] show that this overlapping area is closely correlated with the DM's ambiguous reconstruction of different samples. For instance, due to the large overlap between the intermediate distributions at a certain time step $t_2$, noisy samples drawn from these may be reconstructed ambiguously as either source domain or target domain. This observation leads to a crucial insight: the overlapping area formed through degradation can be interpreted as a domain-shared distribution, which acts as a valuable prior for learning domain-invariant representations. This hypothesis is further substantiated by our quantitative analysis presented in Appendix H, where we empirically validate the correlation between degradation level and domain alignment effectiveness.

It is non-trivial to interpret image degradation as a form of domain bridging, especially considering its fundamental differences from consistency regularization based on data augmentation [40, 93]. This perspective introduces two major challenges: (1) **Wide Range of Degradation Levels:** Degradation operations, when applied incrementally from mild to severe, can gradually eliminate domain-specific attributes. To effectively capture domain-invariant information under such conditions, the encoder of the segmentation network should maintain stable and consistent feature representations across varying levels of degradation. (2) **Semantic Shift Due to Feature Corruption:** While degradation helps remove domain-specific cues, it also inevitably affects domain-invariant features. This degradation of essential semantic information hinders the encoder's ability to extract discriminative representations, which in turn compromises the learning process of the segmentation head. This issue is commonly referred to as the semantic shift problem [1, 86].

To this end, we propose an unsupervised domain bridging approach for semantic segmentation, termed DiDA, which constructs intermediate domains via image degradation defined by a forward diffusion process. We introduce the following key components to integrate DiDA into the UDA training pipeline: (1) **Degradation-based Intermediate Domain Construction:** Based on the definition of the forward diffusion process, we construct a sequence of continuous intermediate domains through simple image degradation operations. These intermediate domains are incorporated into the UDA training process to encourage the model to learn more domain-invariant features from the increasingly

overlapping area of domain distributions. (2) **Semantic Shift Compensation:** To mitigate the semantic shift problem introduced during intermediate domain construction, we propose a diffusion encoder, conditioned on a time embedding module. This encoder disentangles the time-specific semantic shift and, through residual connections across multiple feature levels, compensates for the lost discriminative representations in intermediate domains. This ensures better semantic alignment between extracted features and corresponding labels. (3) **Expansion to Arbitrary Degradation:** To demonstrate the flexibility of our framework, we explore various degradation operations for constructing intermediate domains. The implementation of DiDA is compatible with any image degradation method by simply replacing the Gaussian noise addition in the standard diffusion process. This characteristic showcases the generality and extensibility of our framework. Our approach can be regarded as a plug-and-play training strategy, which can be seamlessly integrated with various UDA methods and network architectures, consistently yielding performance improvements.

In this work, our contributions can be summarized as follows: (1) We propose a novel domain bridging mechanism based on image degradation to facilitate the learning of domain-invariant features. This introduces a new perspective for domain-adaptive semantic segmentation. (2) We design a unified framework that integrates diffusion strategies into the training pipeline of UDA. By introducing a series of degradation-based intermediate domains, our approach enables progressive learning of domain-invariant representations and effectively mitigates the semantic shift problem commonly observed in intermediate domains. (3) We validate the effectiveness and versatility of our approach through extensive experiments on multiple UDA methods, benchmarks, and network architectures. DiDA consistently achieves significant performance improvements across all settings.

## 2 Related Work

### 2.1 Unsupervised Domain Adaptation (UDA)

Unsupervised Domain Adaptation (UDA) aims to transfer semantic knowledge from labeled source domains to unlabeled target domains. Given widespread domain gaps [88, 67], UDA has been extensively studied across various vision tasks, including image classification [29, 28, 57, 35, 34], object detection [11, 12, 51], and semantic segmentation [83, 101, 98, 9]. Semantic segmentation requires assigning a label to each pixel, which often leads to high annotation costs [13, 52, 87]. To address this, various label-efficient methods have been developed, such as semi-supervised learning [79, 62, 78], few-shot learning [54, 59, 55, 53], and domain adaptation [49, 15, 14]. Among them, UDA is particularly valuable, as it eliminates the need for costly pixel-level annotations in new domains. Recent UDA methods for semantic segmentation fall into two main categories: adversarial training and self-training. Adversarial methods align source and target feature distributions via a min-max game between a feature extractor and a domain discriminator [81, 83]. Self-training methods, gaining traction due to the domain-robustness of Transformers [4], adopt a teacher-student framework to generate pseudo labels for target images [82, 38, 60, 50]. Their success hinges on producing reliable pseudo labels through entropy minimization [8], consistency regularization [40], and class-balanced training [48]. Our method builds upon the self-training paradigm, introducing a novel domain bridging mechanism via image degradation to enhance domain invariance and improve adaptation performance.

### 2.2 Generative Models for Segmentation

Generative models like GANs [31] and VAEs [45] map data to latent codes following simple distributions (e.g., Gaussian), enabling data generation and manipulation. Denoising Diffusion Probabilistic Models (DDPMs) [37] extend this by modeling the data-to-latent mapping as a Markov chain with intermediate distributions. To adapt such generative mechanisms to semantic segmentation—a typically discriminative task—some works synthesize paired images and labels to train separate segmentation networks [47, 100], while others directly exploit internal features from generative models [91, 3, 80]. In domain adaptive segmentation, diffusion-based approaches have shown promise. Some methods leverage style transfer [68, 69] or generate diverse domain samples to enhance generalization [66, 99], while others estimate segmentation uncertainty to guide sample selection [26]. In contrast, our method integrates diffusion strategies into the UDA self-training framework, progressively learning domain-invariant representations from images sampled across intermediate distributions.

# 3 Methods

In this section, we first formalize the standard self-training paradigm for UDA and the training process for diffusion models, respectively (Sec.3.1). Then, we introduce theoretical insight (Sec.3.2). After that, we describe our new UDA framework, DiDA, a novel perspective for domain bridging that couples diffusion strategies to improve UDA semantic segmentation performance (Sec.3.3). Finally, we illustrate the details that expand our method to arbitrary choices of image degradation (Sec.3.4).

## 3.1 Preliminary Knowledge

**Self-Training (ST) for UDA.** For domain adaptive semantic segmentation, the source domain can be denoted as $D_s = \{(x_i^S, y_i^S)\}_{i=1}^{N_S}$, where $x_i^S \in X_S$ represents an image with $y_i^S \in Y_S$ as the corresponding pixel-wise one-hot label covering $C$ classes. The target domain can be denoted as $D_t = \{(x_i^T)\}_{i=1}^{N_T}$, which shares the same label space but has no access to target label $Y^T$. In this setting, the supervised loss $\mathcal{L}^S$ can be calculated on the source domain to train a neural network $f_\theta$:

$$\mathcal{L}^S = \sum_{i=1}^{N_S} \mathcal{L}_{ce}(f_\theta(x_i^S), y_i^S, 1), \tag{1}$$

where $\mathcal{L}_{ce}$ denotes the pixel-wise cross-entropy loss:

$$\mathcal{L}_{ce}(\hat{y}_i, y_i, q_i) = -\sum_{j=1}^{H \times W} \sum_{c=1}^{C} q_{(i,j,c)} y_{(i,j,c)} \log \hat{y}_{(i,j,c)}. \tag{2}$$

Self-training introduces a teacher-student framework to generate pseudo-labels $p^T$ for the target domain (see Fig. 2): $p_{(i,j,c)}^T = [c = \text{argmax}_{c'} f_\phi(x_i^S)_{(j,c')}]$, where $f_\phi$ is the teacher network. Then, the pseudo-labels are used to train the network $f_\theta$ on the target domain with the adaptation loss $\mathcal{L}^T$:

$$\mathcal{L}^T = \sum_{i=1}^{N_T} \mathcal{L}_{ce}(f_\theta(x_i^T), p_i^T, q^T). \tag{3}$$

The quality of pseudo-labels is weighted by a confidence estimate $q^T$ [38], which gradually strengthens with increasing accuracy of models. After each training step, the teacher model $f_\phi$ is updated with the exponentially moving average of the weights of $f_\theta$. The segmentation model, $f_\theta$, can be defined as $f_\theta = h \circ g$, where $g : \mathcal{X} \to \mathcal{Z}$ is an encoder that lifts each pixel of the input image in $\mathcal{X}$ to the feature space $\mathcal{Z}$, and $h : \mathcal{Z} \to \mathbb{R}^C$ is a segmentation head which can be viewed as a pixel-wise classifier to give a score for each class.

**Diffusion Model.** Diffusion models learn a series of state transitions to generate high-quality sample from the noise, defined as a forward diffusion process during the training phase. The diffusion process generates intermediate state $x_t$ with a random uniformly sampled t from $\{1,...,T\}$:

$$q(x_t|x_0) = \mathcal{N}(x_t; \sqrt{\bar{\alpha}_t}x_0, (1 - \bar{\alpha}_t)\mathbf{I}), \tag{4}$$

where $\bar{\alpha}_t$ originates from a predefined noise schedule (decreases from 1 to 0). This process can be rewritten through a reparameterization trick:

$$x_t \triangleq \sqrt{\bar{\alpha}_t}x_0 + \sqrt{1 - \bar{\alpha}_t}\epsilon, \quad \epsilon \sim \mathcal{N}(0, \mathbf{I}). \tag{5}$$

Afterword, a network is trained to predict noise $\epsilon$ (or predict sample data $x_0$ directly) from $x_t$, with a reconstruction loss:

$$\mathcal{L}^R = ||f_\theta(x_t, t) - \epsilon||_2^2. \tag{6}$$

## 3.2 Theoretical Insight

In the forward diffusion process, the loss of attributes with different levels of granularity is directly related to the time step. Specifically, we restate following proposition based on the findings in [95]:

**Proposition (Attribute Loss and Time Step).** 1) For each attribute $Z_i$, there exists a minimum time step $t(Z_i)$ such that $Z_i$ is lost with degree $\tau$ at every $t \in \{t(Z_i), \ldots, T\}$. 2) There exists a set $\{\beta_i\}_{i=1}^T$ such that $t(Z_i) > t(Z_j)$ whenever the distribution of $\|x_0 - g_i \cdot x_0\|$ first-order stochastically dominates that of $\|x_0 - g_j \cdot x_0\|$, where $x_0 \sim \mathcal{X}$ uniformly.

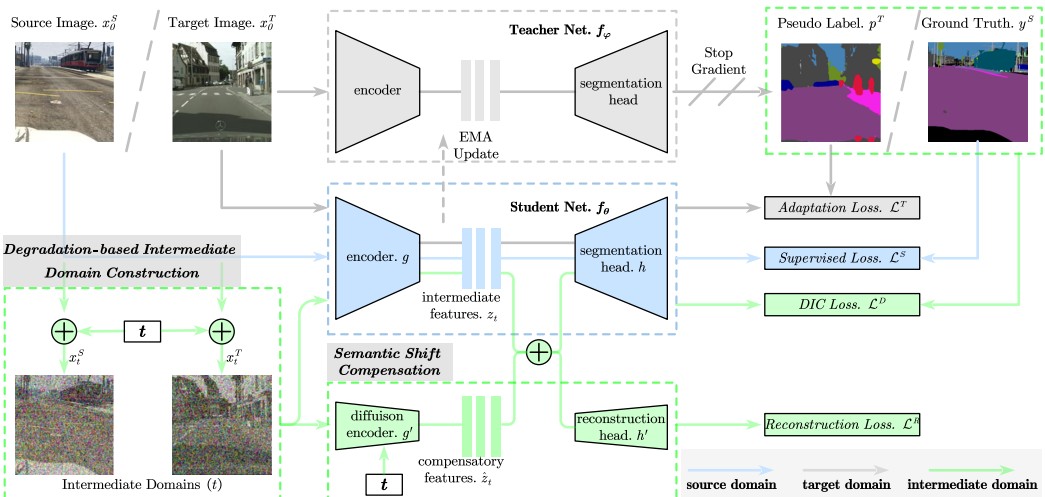

Figure 2: Overview of DiDA framework. We integrate diffusion strategies (green box) with a standard self-training paradigm. While regular frameworks train networks using supervised loss on source domain and unsupervised adaptation loss on target domain, DiDA introduces degradation-based intermediate domains and addresses semantic shift through a diffusion encoder and reconstruction head, which are enabled by degraded image consistency (DIC) loss and reconstruction loss.

The first part of the proposition indicates that once an attribute $Z_i$ is lost at a certain time step $t(Z_i)$ with a specified degree $\tau$, it cannot be recovered in subsequent steps. The second part implies that if the i-th modular attribute transformation $g_i$ induces larger changes in pixel space than $g_j$, then the corresponding attribute $Z_i$ (typically a coarse-grained attribute) is lost at a later time step than $Z_j$ (a fine-grained attribute). This theoretical result reveals that in the forward diffusion process, fine-grained attributes (e.g., texture) are lost earlier than coarse-grained ones (e.g., shape), with granularity measured by the magnitude of pixel-level changes induced by modifying the attribute.

This proposition also shows the overlapping area between domains is closely related to the DM's inherent ambiguity in reconstructing degraded samples. Based on this insight, we reinterpret the overlapping area—created by predefined image degradation—as a domain-shared distribution. This serves as a valuable prior, enabling the network to extract domain-invariant representations.

### 3.3 DiDA Framework

Although intermediate domains generated via image degradation can facilitate cross-domain adaptation, they should contend with a wide range of degradation levels and alleviate the risk of semantic shift. To overcome these challenges, we propose DiDA with two key modules: **Degradation-based Intermediate Domain Construction** and **Semantic Shift Compensation**. Our framework processes images at varying degradation levels and disentangles semantic shift through a dedicated diffusion encoder and reconstruction head. Designed as a general and flexible solution, DiDA seamlessly integrates with existing network architectures and UDA methods.

**Degradation-based Intermediate Domain Construction.** Degradation operations, when applied incrementally from mild to severe, can gradually enlarge overlapping area between domain distributions and eliminate domain-specific attributes. To effectively capture domain-invariant information under such conditions, the encoder $g$ should maintain stable and consistent feature representations across varying levels of degradation. We propose a general approach by formalizing continuous degradation operations as a diffusion forward process.

The learning objective of generative models can be framed as finding a transport map $\mathcal{T} : \mathbb{R}^d \to \mathbb{R}^d$ between two distributions, i.e., $X = \mathcal{T}(Z)$, where $X \sim \pi_x, Z \sim \pi_z$. Typically, $Z$ follows a simple elementary (Gaussian) distribution for sampling to generate $X$ from the data distribution $\pi_x$. In the diffusion process, the transport map is formulated as a Markov chain with learned Gaussian transitions starting at $X_T \sim \pi_z$:

$$X_T \to X_{T-1} \cdots X_t \xrightarrow{p_\theta(X_{t-1}|X_t)} X_{t-1} \cdots \to X_0, \tag{7}$$

which is termed as a reverse process in contrast to equation (4). In our case, $X_0 \sim \{\pi_s, \pi_t\}$ represents the distribution of the source or target domain, respectively. Since $X_1, X_2, ..., X_T$ can be viewed as latent codes of the same dimensionality as the data $X_0$, we consider them as intermediate domains, which possess gradually increasing overlapping area compared to the original $X_0$. This formulation enables a degradation-based domain bridging mechanism within the diffusion framework.

**Semantic Shift Compensation.** While degradation helps remove domain-specific cues, it also inevitably affects domain-invariant features. This degradation of essential semantic information hinders the encoder's ability to extract discriminative representations, known as semantic shift. To address this challenge, we propose a compensation mechanism that effectively disentangles the semantic shift and aligns the extracted features with label semantics throughout the degradation process, enabling the network to process intermediate domains at any degradation level while maintaining semantic consistency.

In this module, a trainable diffusion encoder, $g'$, is introduced to map each degraded image $x_t$ to the feature space $\hat{\mathcal{Z}}$ given $t$. This diffusion encoder $g'$ is designed to capture the semantic shift information of the segmentation network's encoder $g$ when taking $x_t$ as input. To this end, a time embedding module is added to specify the diffusion time $t$. It is implemented by the Transformer sinusoidal position embedding [84] to condition all blocks of $g'$ on $t$. Then, the internal feature $z'_{(t,i)}$ in $block'_i$ is modulated with shift and bias:

$$\hat{z}_{(t,i)} = z'_{(t,i)}(MLP_s^i \circ Embed(t) + 1) + MLP_b^i \circ Embed(t), \tag{8}$$

which is operated at the channel dimension of $z'_{(t,i)}$. The resulting feature $\hat{z}_t$ is required to compensate for the intermediate feature $z_t$ from $g$ to minimize the reconstruction error by being supervised with the loss $L^R$ (see equation (6)), where $f_\theta$ is replaced by $\hat{f}_\theta = h' \circ (g + g')$. We perform feature fusion by adding hierarchical features through residual connections. Through this module, the network is empowered to precisely disentangle the degree of perturbation and align the extracted features with label semantics by compensating for the semantic shift. Meanwhile, it retains the capability to adapt domain-invariant knowledge to the original domain distribution.

To leverage intermediate domains for improved adaptation, we introduce a Degraded Image Consistency (DIC) loss:

$$\mathcal{L}^D = \sum_{i=1}^{N_S} \mathcal{L}_{ce}(\bar{f}_\theta(x_{i,t}^S, t), y_i^S, 1) + \sum_{i=1}^{N_T} \mathcal{L}_{ce}(\bar{f}_\theta(x_{i,t}^T, t), p_i^T, q^T), \tag{9}$$

where $\bar{f}_\theta = h \circ (g + g')$ and $x_{i,t}$ is degrade image of $x_{i,0}$, which can be obtained with equation (5). This loss enforces consistency between predictions on degraded and original images.

**Training and Inference.** The training pipeline is shown in Fig. 2 and detailed in Appendix A. The original operation in the self-training framework is fully retained, and DiDA can be considered as an additional plugin. In each training iteration, we conduct the forward process with a fixed noise schedule $\bar{\alpha}_t$ and random $t$ sampled from a uniform distribution between 1 and $T$ on the current training batch. Then, they are fed to the network $\bar{f}_\theta / \hat{f}_\theta$, which shares the same weights with student net $f_\theta$, and diffusion time $t$ is encoded through the time embedding module in diffusion encoder $g'$. The outputs of this step, prediction for segmentation map and noise, are individually supervised by DIC loss and reconstruction loss. The overall loss $\mathcal{L}$ for DiDA is the weighted sum of the presented loss components:

$$\mathcal{L} = \mathcal{L}^S + \mathcal{L}^T + \lambda_D \mathcal{L}^D + \lambda_R \mathcal{L}^R. \tag{10}$$

During regular inference, only the backbone segmentation network $f_\theta = h \circ g$ is used, while the diffusion-specific components—$g'$ and $h'$—are entirely removed. The input is an unprocessed image from $X_0$, and there is no need to use the diffusion encoder or reconstruction head for noise prediction, meaning that no additional time consumption or network structure changes are required in this stage.

### 3.4 Expansion to Arbitrary Degradation

Diffusion models demonstrate flexibility in their choice of degradation operations beyond traditional Gaussian noise [2]. Building on this insight, we extend DiDA to support various degradation types while maintaining our core principles of intermediate domain construction and semantic shift perception. To demonstrate this generalization, we implement forward diffusion processes based

Table 1: Quantitative results of DiDA on different methods and benchmarks with CNN-based model (C) or Transformer-based model (T). ∗ denotes the reproduced result.

| Method | GTA.→CS. | | SYN.→CS. | | CS.→ACDC |
|---|---|---|---|---|---|
| | C | T | C | T | T |
| DAFormer [38] | 56.0 | 68.3 | 54.7 | 60.9 | 55.4 |
| +DiDA | $58.3_{\uparrow 2.3}$ | $70.3_{\uparrow 2.0}$ | $57.6_{\uparrow 2.9}$ | $63.1_{\uparrow 2.2}$ | $59.1_{\uparrow 3.7}$ |
| HRDA [39] | 63.0 | 73.8 | 61.2 | 65.8 | 68.0 |
| +DiDA | $64.3_{\uparrow 1.3}$ | $75.4_{\uparrow 1.6}$ | $62.6_{\uparrow 1.4}$ | $67.8_{\uparrow 2.0}$ | $70.7_{\uparrow 2.7}$ |
| MIC [40] | 64.2 | 75.5* | 62.4* | 67.3 | 69.8* |
| +DiDA | $65.0_{\uparrow 0.8}$ | $\mathbf{76.8}_{\uparrow 1.3}$ | $63.5_{\uparrow 1.1}$ | $\mathbf{68.6}_{\uparrow 1.3}$ | $\mathbf{72.1}_{\uparrow 2.3}$ |

Table 2: Component ablation analysis of DiDA built with DAFormer on GTA.→CS.(val).

| $\mathcal{L}^D$ | $\mathcal{L}^R$ | $g_{time}$ | $g'$ | $h'$ | mIoU |
|---|---|---|---|---|---|
| - | - | - | - | - | 68.3 |
| ✓ | - | - | - | - | 66.5 |
| ✓ | - | ✓ | - | - | 69.5 |
| - | ✓ | ✓ | - | ✓ | 67.9 |
| ✓ | ✓ | ✓ | - | - | 69.4 |
| ✓ | ✓ | ✓ | - | - | 69.9 |
| ✓ | ✓ | - | ✓ | ✓ | **70.3** |

on two fundamental vision tasks: deblurring and inpainting. These implementations preserve the network's ability to perceive degradation levels and compensate for semantic shifts across different degradation operations. Please refer to Appendix B for details on the implementation.

# 4 Experiments

## 4.1 Implementation Details

**Datasets.** To comprehensively evaluate the performance of DiDA, we follow standard UDA protocols and conduct experiments on both synthetic-to-real and clear-to-adverse-weather adaptation scenarios. As synthetic datasets, we use GTAv [70] containing 24,966 images and SYNTHIA [71] with 9,400 images. For real-world datasets, we employ Cityscapes [21] with 2,975 training and 500 validation images representing clear weather conditions, and ACDC [72] containing 1,600 training, 406 validation, and 2,000 test images capturing adverse weather conditions (fog, night, rain, and snow). We report the intersection over union for each class as well as the mean IoU over all classes.

**Base Segmentation Architectures and UDA Methods** To demonstrate the versatility and adaptability of our method, we implement it on two widely-used network architectures and three progressively enhanced baseline methods. Specifically, we employ DeepLabV2 [5] with ResNet-101 [32] backbone and DAFormer [38] with MiT-B5 [90] backbone. Both of these network architectures have been pretrained on the ImageNet-1k [24] dataset. Building upon this foundation, we apply our framework to different methods in the DAFormer series, including DAFormer [38], HRDA [39], and MIC [40].

**Training Details.** We implement DiDA based on the MMSegmentation [20] framework. Depending on the complexity of the network architectures and UDA frameworks, all experiments are conducted on one or two RTX-3090 GPUs with 24 GB memory, with 40K training iterations and a batch size of 2. We train the network using the AdamW optimizer, with learning rates of $6 \times 10^{-5}$ for the encoder and $6 \times 10^{-4}$ for the decoder, a weight decay of 0.01, and a linear learning rate warm-up strategy for the first 1.5K iterations. The EMA coefficient for updating the teacher network is set to 0.999. We set $T = 100$ and use a sigmoid schedule [44] to obtain $\bar{\alpha}_t$. To achieve scale and quantity matching during feature fusion, we initialize a diffusion encoder $h'$ with the same structure as the corresponding the segmentation encoder, along with extra time embedding modules. For the reconstruction head $g'$, we initialize an ASPP module [5] with a linear projector. For the reconstruction loss, the clean image $x_0$ or sampled noise $\epsilon$ is downsampled at a rate of $4\times$ or $8\times$ to match the input of the reconstruction head. We set DIC loss weight $\lambda^D$ to 0.5, and reconstruction loss weight $\lambda^R$ to 5 for DAFormer and 1 for DeepLabV2 to ensure a similar gradient magnitude induced by these different components. Finally, we report the mIoU using the last checkpoint of the student model $f_\theta$ without model selection.

## 4.2 Evaluation on Benchmark Datasets

**Overall Quantitative Results.** Tab. 1 shows quantitative results by building DiDA upon three mainstream methods with different architectures. We report results based on whole inference on DAFormer and slide inference on HRDA and MIC without any other test time augmentation strategies for a fair comparison. Whether with Transformer-based or CNN-based backbones, DiDA achieves consistent gains beyond all baselines, ranging from 0.8% to 3.7%. As expected, the performance improvement generally decreases when the baseline is stronger and closer to performance saturation. When implemented with DeepLabV2, the performance gain is slightly lower than DAFormer due to the coarse output ($8\times$ downsampling) for the optimization goal of reconstruction. It is worth noting that DiDA achieves new state-of-the-art performance when applied on MIC.

Table 3: UDA segmentation performance on GTA.→CS., where the improvement by DiDA is marked as **bold**. The results are acquired with CNN-based model (C) in the first group or Transformer-based model (T) in the second group. ∗ denotes the reproduced result.

| Method | Arch. | Road | Sidewalk | Building | Wall | Fence | Pole | Light | Sign | Veg | Terrain | Sky | Person | Rider | Car | Truck | Bus | Train | Motor | Bike | mIoU |
|---|---|---|---|---|---|---|---|---|---|---|---|---|---|---|---|---|---|---|---|---|---|
| DACS [82] | C | 89.9 | 39.7 | 87.9 | 39.7 | 39.5 | 38.5 | 46.4 | 52.8 | 88.0 | 44.0 | 88.8 | 67.2 | 35.8 | 84.5 | 45.7 | 50.2 | 0.2 | 27.3 | 34.0 | 52.1 |
| I2F [61] | C | 90.8 | 48.7 | 85.2 | 30.6 | 28.0 | 33.3 | 46.4 | 40.0 | 85.6 | 39.1 | 88.1 | 61.8 | 35.0 | 86.7 | 46.3 | 55.6 | 11.6 | 44.7 | 54.3 | 53.3 |
| ProDA [98] | C | 87.8 | 56.0 | 79.7 | 46.3 | 44.8 | 45.6 | 53.5 | 53.5 | 88.6 | 45.2 | 82.1 | 70.7 | 39.2 | 88.8 | 45.5 | 50.4 | 1.0 | 48.9 | 56.4 | 57.5 |
| DAP [42] | C | 94.5 | 63.1 | 89.1 | 29.8 | 47.5 | 50.4 | 56.7 | 58.7 | 89.5 | 50.2 | 87.0 | 73.6 | 38.6 | 91.3 | 50.2 | 52.9 | 0.0 | 50.2 | 63.5 | 59.8 |
| CPSL [48] | C | 92.3 | 59.5 | 84.9 | 45.7 | 29.7 | 52.8 | 61.5 | 59.5 | 87.9 | 41.6 | 85.0 | 73.0 | 35.5 | 90.4 | 48.7 | 73.9 | 26.3 | 53.8 | 53.9 | 60.8 |
| MIC [40] | C | 96.5 | 74.3 | 90.4 | 47.1 | 42.8 | 50.3 | 61.7 | 62.3 | 90.3 | 49.2 | 90.7 | 77.8 | 53.2 | 93.0 | 66.2 | 68.0 | 6.8 | 38.0 | 60.6 | 64.2 |
| +DiDA | C | 96.6 | 74.6 | 89.2 | **47.5** | **44.2** | 50.0 | 61.2 | 60.6 | 90.4 | **51.9** | **91.8** | 76.5 | **53.8** | **93.5** | **67.1** | 63.7 | 5.8 | **50.0** | **66.7** | **65.0** |
| TransDA [10] | T | 94.7 | 64.2 | 89.2 | 48.1 | 45.8 | 50.1 | 60.2 | 40.8 | 90.4 | 50.2 | 93.7 | 76.7 | 47.6 | 92.5 | 56.8 | 60.1 | 47.6 | 49.6 | 55.4 | 63.9 |
| ADFormer [33] | T | 96.7 | 75.1 | 88.8 | 57.5 | 45.9 | 45.6 | 55.4 | 59.8 | 90.2 | 45.6 | 92.1 | 70.8 | 43.0 | 91.0 | 78.9 | 79.3 | 68.7 | 52.7 | 65.0 | 69.2 |
| CoPT [63] | T | 97.6 | 80.9 | 91.6 | 62.1 | 55.9 | 59.3 | 66.7 | 70.5 | 91.9 | 53.0 | 94.4 | 80.0 | 55.6 | 94.7 | 87.1 | 88.6 | 82.1 | 65.0 | 68.8 | 76.1 |
| DAFormer [38] | T | 95.7 | 70.2 | 89.4 | 53.5 | 48.1 | 49.6 | 55.8 | 59.4 | 89.9 | 47.9 | 92.5 | 72.2 | 44.7 | 92.3 | 74.5 | 78.2 | 65.1 | 55.9 | 61.8 | 68.3 |
| +FST [25] | T | 95.3 | 67.7 | 89.3 | 55.5 | 47.1 | 50.1 | 57.2 | 58.6 | 89.9 | 51.0 | 92.9 | 72.7 | 46.3 | 92.5 | 78.0 | 81.6 | 74.4 | 57.7 | 62.6 | 69.3 |
| +DiDA(B) | T | **97.2** | **76.3** | 89.2 | **58.0** | **51.1** | **53.6** | 57.5 | **63.5** | 90.0 | **51.3** | 92.3 | 71.7 | 43.8 | 92.1 | 69.2 | **81.4** | **72.1** | 56.0 | **64.6** | **70.0** |
| +DiDA(M) | T | 96.4 | 75.3 | 90.5 | 57.6 | 49.2 | 53.4 | 58.6 | 64.4 | 90.5 | 52.6 | 92.5 | 71.8 | 40.8 | 92.6 | 70.6 | 81.7 | 66.1 | 57.6 | 64.0 | 69.8 |
| +DiDA | T | 96.9 | 74.7 | 88.9 | 54.4 | 49.8 | 53.5 | 57.5 | 63.9 | 90.6 | 50.4 | 92.2 | 71.5 | 50.8 | 92.2 | 76.1 | 82.1 | 70.7 | 53.2 | 66.8 | 70.3 |
| HRDA [39] | T | 96.4 | 74.4 | 91.0 | 61.6 | 51.5 | 57.1 | 63.9 | 69.3 | 91.3 | 48.4 | 94.2 | 79.0 | 52.9 | 93.9 | 84.1 | 85.7 | 75.9 | 63.9 | 67.5 | 73.8 |
| +DiGA [75] | T | 97.0 | 78.6 | 91.3 | 60.8 | 56.7 | 56.5 | 64.4 | 69.9 | 91.5 | 50.8 | 93.7 | 79.2 | 55.2 | 93.7 | 78.3 | 86.9 | 77.8 | 63.7 | 65.8 | 74.3 |
| +DiDA | T | **97.4** | **79.6** | **91.6** | **62.9** | 55.7 | **59.2** | **68.0** | **70.3** | **92.0** | **55.5** | 93.8 | **80.4** | 52.5 | **94.8** | **86.9** | **87.0** | 69.3 | **66.2** | **68.9** | **75.4** |
| MIC* [40] | T | 97.4 | 80.1 | 91.7 | 61.4 | 56.9 | 60.3 | 66.4 | 71.3 | 91.7 | 51.2 | 94.1 | 79.8 | 55.6 | 94.6 | 85.9 | 88.5 | 74.3 | 64.7 | 68.1 | 75.5 |
| +DTS [43] | T | 97.0 | 80.4 | 91.8 | 60.6 | 58.7 | 61.7 | 7.9 | 73.2 | 92.0 | 45.4 | 94.3 | 81.3 | 58.7 | 95.0 | 87.9 | 90.7 | 82.2 | 65.7 | 69.0 | 76.5 |
| +DiDA | T | **97.9** | **81.0** | **92.4** | **62.0** | 57.7 | 60.5 | 63.2 | **76.6** | **92.3** | **56.4** | **94.4** | 79.2 | 54.4 | 94.7 | 86.2 | 90.4 | 81.8 | 65.8 | **71.6** | **76.8** |

**Class-level Comparison.** We further display the class-wise performance on each benchmark in Tab. 3 and Tab. 4, with additional results provided in Tab. 5 (Appendix D), for detailed comparison. When combined with DiDA, most of classes achieve higher accuracy. We investigate that performance on *road*, *sidewalk*, *fence*, and *terrain* achieves consistent and relatively significant improvements over all UDA methods and datasets. These classes constitute the main scene and comprise abundant domain-specific texture information. By adding random Gaussian noise to images, these textures are broken while the context information is preserved, which is more robust over domains. Since DiDA establishes domain bridging through these progressively intensifying degraded images between domains, the network can focus more on domain-invariant context to enhance the adapting ability. Furthermore, we implement the extended version of DiDA on the DAFormer baseline with **Blur** and **Mask**, denoted as **B** and **M**, respectively. These extensions obtain less gain than the default version but show a certain potential if well-designed.

**Comparison with other plug-in methods.** We further compare DiDA with other plug-in approaches, such as FST [25], DiGA [75], and DTS [43]. Our method consistently demonstrates superior performance compared to these methods when built with the same baselines.

### 4.3 Diagnostic Experiments

Please refer to the Appendix D-L for further analysis, where we provide more experiment results, deeper ablation studies, and more visualization.

**Component Ablation Analysis.** To gain deeper insights, we analyze the proposed approach by ablating the components and evaluating the performance with DAFormer as the baseline on GTA→CS. The results are presented in Tab. 2. The DiDA with complete components achieves a 2.0 mIoU gain on this baseline. To analyze the diffusion encoder and the module of time embedding respectively, we use the segmentation encoder $g$ directly conditioned by time embedding as $g_{time}$ without an extra diffusion encoder, as the baseline for ablation studies (row 6). We first ablate the module of time embedding, implying that the original network is equivalently trained with the randomly degraded images as additional data augmentation. The performance decreases heavily in this case (row 2) due to the excessive degradation of data, known as semantic shift problem. The introduced time embedding plays a vital role in encoding noise intensity information, indicating that we alleviate this issue by encoding time-specific semantic shift information. Then, we discard the loss terms, i.e., $\mathcal{L}^D$ and $\mathcal{L}^R$, respectively (rows 3 and 4). Without $\mathcal{L}^R$, DiDA obtains less improvement, although it does not constrain segmentation results directly, indicating that $\mathcal{L}^R$ helps to learn the module of time embedding and perceive the semantic shift. Discarding $\mathcal{L}^D$ brings an accuracy drop since introducing $\mathcal{L}^R$ only is inconsistent with the learning objective for high-level semantics. After that, the extra reconstruction head $h'$ is ablated (row 5), which means that we obtain reconstruction results from the segmentation head $h$. It is also essential to avoid excessive disturbance for originally learned

Table 4: UDA segmentation performance on SYN.→CS., where the improvement by DiDA is marked as **bold**. The results are acquired with CNN-based model (C) in the first group or Transformer-based model (T) in the second group.

| Method | Arch. | Road | Sidewalk | Building | Wall | Fence | Pole | Light | Sign | Veg | Terrain | Sky | Person | Rider | Car | Truck | Bus | Train | Motor | Bike | mIoU |
|---|---|---|---|---|---|---|---|---|---|---|---|---|---|---|---|---|---|---|---|---|---|
| DACS [82] | C | 80.6 | 25.1 | 81.9 | 21.5 | 2.9 | 37.2 | 22.7 | 24.0 | 83.7 | - | 90.8 | 67.6 | 38.3 | 82.9 | - | 38.9 | - | 28.5 | 47.6 | 48.3 |
| I2F [61] | C | 84.9 | 44.7 | 82.2 | 9.1 | 1.9 | 36.2 | 42.1 | 40.2 | 83.8 | - | 84.2 | 68.9 | 35.3 | 83.0 | - | 49.8 | - | 30.1 | 52.4 | 51.8 |
| ProDA [98] | C | 87.8 | 45.7 | 84.6 | 37.1 | 0.6 | 44.0 | 54.6 | 37.0 | 88.1 | - | 84.4 | 74.2 | 24.3 | 88.2 | - | 51.1 | - | 40.5 | 45.6 | 55.5 |
| DAP [42] | C | 84.2 | 46.5 | 82.5 | 35.1 | 0.2 | 46.7 | 53.6 | 45.7 | 89.3 | - | 87.5 | 75.7 | 34.6 | 91.7 | - | 73.5 | - | 49.4 | 60.5 | 59.8 |
| CPSL [48] | C | 87.2 | 43.9 | 85.5 | 33.6 | 0.3 | 47.7 | 57.4 | 37.2 | 87.8 | - | 88.5 | 79.0 | 32.0 | 90.6 | - | 49.4 | - | 50.8 | 59.8 | 57.9 |
| TransDA [10] | T | 90.4 | 54.8 | 86.4 | 31.1 | 1.7 | 53.8 | 61.1 | 37.1 | 90.3 | - | 93.0 | 71.2 | 25.3 | 92.3 | - | 66.0 | - | 44.4 | 49.8 | 59.3 |
| ADFormer [33] | T | 91.8 | 53.6 | 87.0 | 40.5 | 5.2 | 46.8 | 52.1 | 54.9 | 88.4 | - | 92.6 | 72.5 | 45.7 | 86.1 | - | 61.6 | - | 50.4 | 64.4 | 62.1 |
| CoPT [63] | T | 83.4 | 44.3 | 90.0 | 50.4 | 8.0 | 60.0 | 67.0 | 63.0 | 87.5 | - | 94.8 | 81.1 | 58.6 | 89.7 | - | 66.5 | - | 68.9 | 65.0 | 67.4 |
| DAFormer [38] | T | 84.5 | 40.7 | 88.4 | 41.5 | 6.5 | 50.0 | 55.0 | 54.6 | 86.0 | - | 89.8 | 73.2 | 48.2 | 87.2 | - | 53.2 | - | 53.9 | 61.7 | 60.9 |
| +FST [25] | T | 88.3 | 46.1 | 88.0 | 41.7 | 7.3 | 50.1 | 53.6 | 52.5 | 87.4 | - | 91.5 | 73.9 | 48.1 | 85.3 | - | 58.6 | - | 55.9 | 63.4 | 61.9 |
| **+DiDA** | T | **87.8** | **47.5** | **88.9** | **43.1** | **9.8** | **51.6** | **56.8** | **56.1** | **86.5** | - | **90.1** | **76.1** | **46.5** | **88.8** | - | **56.8** | - | **59.3** | **63.2** | **63.1** |
| HRDA [39] | T | 85.2 | 47.7 | 88.8 | 49.5 | 4.8 | 57.2 | 65.7 | 60.9 | 85.3 | - | 92.9 | 79.4 | 52.8 | 89.0 | - | 64.7 | - | 63.9 | 64.9 | 65.8 |
| +DiGA [75] | T | 88.5 | 49.9 | 90.1 | 51.4 | 6.6 | 55.3 | 64.8 | 62.7 | 88.2 | - | 93.5 | 78.6 | 51.8 | 89.5 | - | 62.2 | - | 61.0 | 65.8 | 66.2 |
| **+DiDA** | T | **87.9** | **52.9** | **89.6** | **54.3** | **11.6** | **56.6** | **63.8** | **61.2** | **87.6** | - | **94.1** | **79.9** | **54.2** | **90.5** | - | **71.5** | - | **67.1** | **62.3** | **67.8** |
| MIC [40] | T | 86.6 | 50.5 | 89.3 | 47.9 | 7.8 | 59.4 | 66.7 | 63.4 | 87.1 | - | 94.6 | 81.0 | 58.9 | 90.1 | - | 61.9 | - | 67.1 | 64.3 | 67.3 |
| +DTS [43] | T | 89.1 | 54.9 | 89.0 | 39.1 | 8.7 | 61.6 | 67.4 | 64.3 | 88.8 | - | 94.0 | 82.2 | 60.7 | 89.6 | - | 62.6 | - | 68.5 | 64.9 | 67.8 |
| **+DiDA** | T | **89.2** | **55.9** | **89.6** | **49.0** | **8.4** | **58.9** | **66.1** | **64.4** | **88.5** | - | **94.6** | **80.6** | **59.4** | **89.2** | - | **69.2** | - | **66.7** | **68.4** | **68.6** |

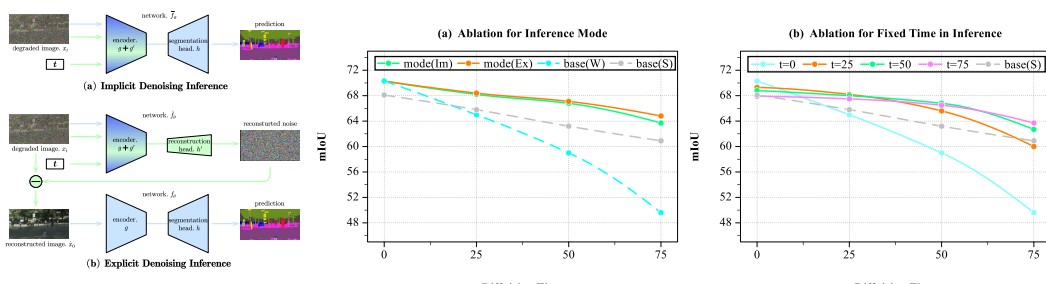

Figure 3: Demonstration of two modes of inference.

Figure 4: The performance variation with the degraded level.

features. In the end, the extra diffusion encoder is ablated (rows 6 and 7). Although $g_{time}$ can perceive the semantic shift implicitly enabled by time embedding, it is more effective to introduce an extra diffusion encoder to compensate for the lost discriminative representation.

**How DiDA Works.** To further comprehend the working mechanism of DiDA, we design two modes for inference on degraded images with known diffusion time $t$, shown in Fig. 3. The first mode, called implicit denoising inference, is the same as what we used in the training phase of DiDA. The network $\bar{f}_\theta$ takes degraded images and $t$ as inputs and generates segmentation results immediately. In contrast with this mode, the second mode segments the degraded images indirectly by predicting the noise through $f_\theta$ first for reconstruction and feeding them back to the network $\bar{f}_\theta$, which shares the same weight with $\bar{f}_\theta$ but no diffusion encoder, to obtain the final prediction. This mode is termed explicit denoising inference. We evaluate the performance difference between the two modes on different noise levels and construct two baselines for comparison. The strong baseline, called base(S), trains and inferences each model separately on intermediate domains with different noise levels. For the weak baseline, termed as base(W), we execute inference on these intermediate domains with $f_\theta$. Fig. 4 (a) plots the performance curve throughout the diffusion forward process. The performance with implicit denoising inference is slightly lower than the explicit mode, and the gap between them is tiny. Both modes beat the strong baseline using a single network trained only once. It indicates that DiDA can perceive the semantic shift precisely and extract features in an implicit denoising manner during the training phase. Furthermore, unlike the explicit mode, the network $\bar{f}_\theta$ learns domain-invariant features directly from the degraded images to bridge the domain gap and heighten the adapting ability. These two factors jointly contribute to improving overall performance on UDA segmentation.

**Inference with Fixed Diffusion Time.** Based on the above discussion, we further evaluate the performance for inference with different fixed diffusion times $t$. The results are summarized in Fig. 4 (b). Note that $t = 0$ is equivalent to inference without the diffusion encoder, i.e., $f_\theta$, which is the same as the weak baseline defined above. With the change of fixed diffusion time, there are two common properties among these curves: (i) the performance decreases along with the forward diffusion process, and (ii) at each fixed level of the forward process, inference with the same fixed

diffusion time $t$ can achieve the current best precision, which is in accordance with the intuition. We can deduce that the higher $t$ is fixed in the inference procedure, the flatter the corresponding performance curve will be. Although the optimal performance degrades, this phenomenon reveals a desirable property: inference with a fixed diffusion time $t$ can enhance the network's robustness against input perturbations, thereby improving its anti-interference capability.

## 5 Conclusions

In this paper, we propose DiDA, a degradation-based bridging framework for domain adaptive semantic segmentation. By simulating intermediate domains through simple image degradations and formalizing them as a diffusion process, DiDA effectively mitigates semantic shift and promotes domain-invariant feature learning. The framework is general and modular, supporting various degradation types and seamlessly integrating with diverse UDA methods and backbones. Extensive experiments demonstrate that DiDA consistently improves performance and achieves new state-of-the-art results on multiple standard UDA benchmarks.

## Acknowledgements

This work was partially supported by the National Key R&D Program of China (Grant No. 2024YFB3909902), and the Youth Innovation Promotion Association of the Chinese Academy of Sciences (CAS).

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

# A Implementation Details of DiDA

In this section, we provide the pseudo algorithms to explain implementation details of DiDA, as shown in Alg. 1. Our method is designed as a plug-and-play module applicable to any self-training-based UDA framework, introducing no additional computational overhead during inference.

---

**Algorithm 1** Pseudo algorithms of DiDA.

---

1: **Inputs:** Source Domain $D_s = (x_i^S, y_i^S)_{i=1}^{N_S}$, Target Domain $D_t = (x_i^T)_{i=1}^{N_T}$
2: **Define:** Student Network $f_\theta$, Teacher Network $f_\phi$, Diffusion Encoder $g'$, Reconstruction Head $h'$, Noise Schedule $\bar{\alpha}_t$, Diffusion Steps $T$, Momentum Coefficient $\beta$
3: **Output:** Student Network $f_\theta$
4: **for** each batch of $(x_i^S, y_i^S)$, $x_i^T$ in $D_s$, $D_t$ **do**
5:     *# Source Domain:*
6:     Calculate $\mathcal{L}^S$ for $f_\theta$ by Eq. (1)                                     ▷ *Supervised loss*
7:     *# Target Domain:*
8:     Obtain pseudo-labels from $f_\phi$ by Eq. (3)
9:     Calculate $\mathcal{L}^T$ for $f_\theta$ by Eq. (4)                                     ▷ *Adaptation loss*
10:     *# Degradation-based Intermediate Domain Construction:*
11:     Sample $t \sim \mathrm{Uniform}(1, T)$
12:     Obtain degraded images $x_{i,t}^S, x_{i,t}^T$ by Eq. (6)
13:     *# Semantic Shift Compensation:*
14:     Calculate $\mathcal{L}^D$ for $\hat{f}_\theta = h \circ (g + g')$ by Eq. (10)          ▷ *Degraded image consistency loss*
15:     Calculate $\mathcal{L}^R$ for $\hat{f}_\theta = h' \circ (g + g')$ by Eq. (7)          ▷ *Reconstruction loss*
16:     *# Training:*
17:     Gradient backward $\mathcal{L}^S + \mathcal{L}^T + \lambda_D \mathcal{L}^D + \lambda_R \mathcal{L}^R$          ▷ *Update student model*
18:     *# EMA Update:*
19:     $\phi \leftarrow \beta\phi + (1 - \beta)\theta$                                     ▷ *Update teacher model*
20: **end for**

---

# B Details in Expansion Versions

In this section, we describe the details of our implementation for expansion versions of DiDA, in which we replace the degradation operation in the diffusion forward process with **blur** and **mask**.

**Image Blur.** Given the Gaussian kernels $\{G_s\}$, the forward process can be simply defined as:

$$x_t = G_t * x_{t-1} = G_t * ...G_1 * x_0 = \bar{G}_t * x_0, \tag{11}$$

where * represents the convolution operator to apply the Gaussian blur operation on the image. Then, the model $\hat{f}_\theta$ is trained for deblurring to invert this blurred diffusion process.

Following the setting of cold diffusion [2], we define T Gaussian kernels: $G_1, ..., G_T$ to execute gradual blurring. For instance, we set $T = 100$ with a Gaussian kernel of $31 \times 31$, and the standard deviation of the Gaussian kernel grows exponentially with time $t$ at the rate of 0.02.

**Image Mask.** To implement the incremental mask operation on the image with the diffusion time steps, we define this process with cowmask [27]. With the schedule $\bar{\alpha}_t$ as a threshold, we can generate the cowmask $\mathcal{M}_t$ and obtain $x_t$ by element-wise multiplication of the mask and image:

$$x_t = \mathcal{M}_t \odot x_0, \tag{12}$$

and the inpainting model is trained to restore the image. The procedure for generating a masked image with cowmask and $\tau$ is provided in Algorithm 2, and we wet the std $\delta = 6$.

To extend our method with the above-defined forward process, we only need to replace the degraded image sampling operation and modify the reconstruction loss to predict the clean image $x_0$ directly:

$$\mathcal{L}^R = \lambda_t ||\hat{f}_\theta(x_t, t) - x_0||_2^2, \tag{13}$$

where $\lambda_t$ is a $t$-dependent loss weight, defined as a fixed value computed from the noise schedule $\{\alpha_t\}_{t=1}^T$ [73], introduced to balance the contribution of different degradation levels.

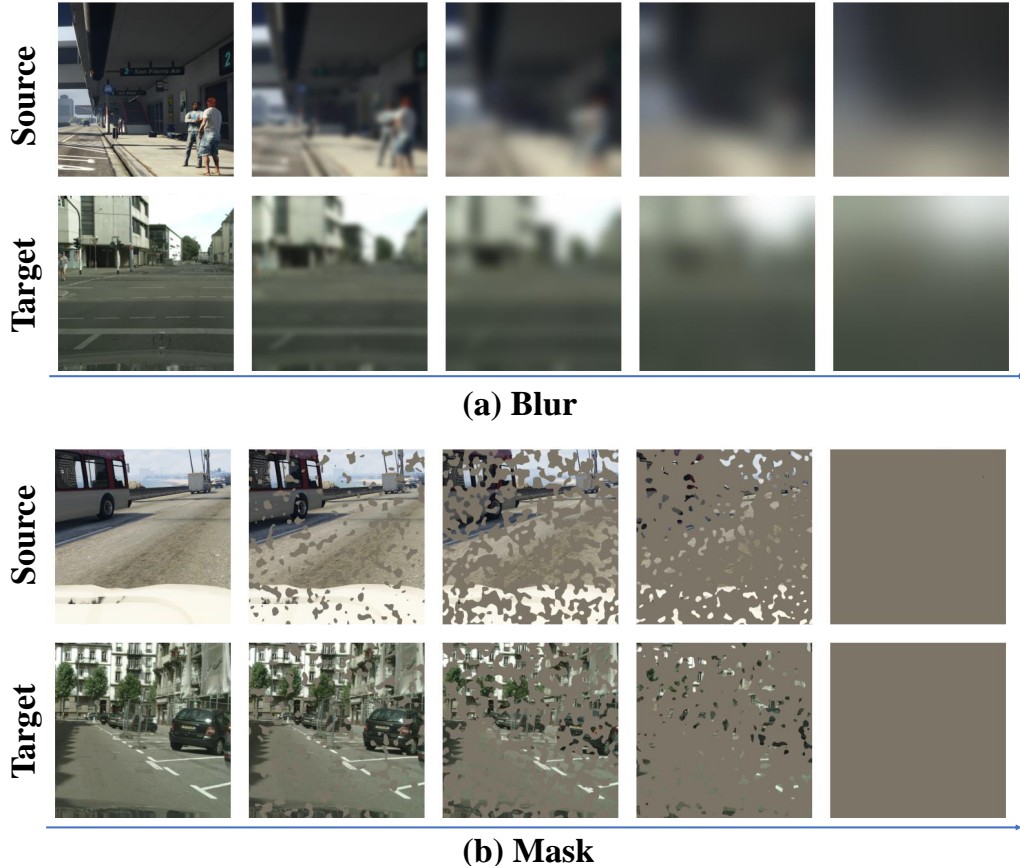

**(a) Blur**

**(b) Mask**

Figure 5: The examples that extend DiDA to any other forward diffusion process defined by arbitrary image degradation operations. (a) shows image blurred process and (b) shows image masked process.

---

**Algorithm 2** CowMask generation algorithm with the threshold $\tau \in [0, 1]$ as the ratio.

---

**Require:** original image $x^O$, threshold $\tau$, std $\delta$
**Ensure:** masked image $x^M$
 1: sample Gaussian noise $\epsilon \sim \mathcal{N}(0, \mathbf{I})$
 2: filter noise $\epsilon_f = gaussian\_filter\_2d(\epsilon, \delta)$
 3: compute mean m $= mean(\epsilon_f)$
 4: compute std_dev s $= std\_dev(\epsilon_f)$
 5: compute noise threshold $p = $ m $+ \sqrt{2}erf^{-1}(2\tau - 1)$s
 6: threshold filtered noise $\mathcal{M} = \epsilon_f < p$
 7: mask image $x^M = x^O \odot \mathcal{M}$
 8: **return** $x^M$

---

## C  Architecture of Time Embedding Module

In this section, we illustrate the details of the time embedding module introduced in different backbones. As shown in Fig. 6, we condition all blocks of models on $t$, whether Transformer-based architecture or CNN-based architecture. The diffusion time $t$ is first encoded by Transformer sinusoidal position embedding [84] and projected to time embedding through 2 layers MLP. Then, the time embedding is encoded by $MLP_s^i$ and $MLP_b^i$ to obtain shift and bias and ensure the same channel dimension as the corresponding feature. In the end, each block's internal feature $f_i$ is modulated through multiplication and addition operations at the channel dimension.

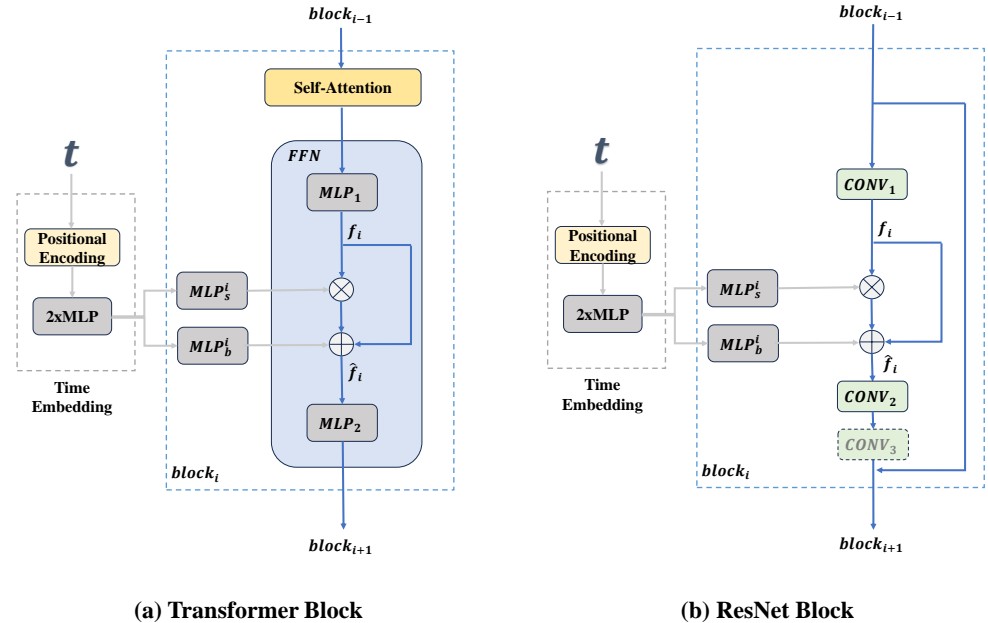

| (a) Transformer Block | (b) ResNet Block |

Figure 6: Architecture of time embedding module in different backbones.

Table 5: UDA segmentation performance on CS.→ACDC, where the improvement by DiDA is marked as **bold**. The results are acquired with CNN-based model (C) in the first group or Transformer-based model (T) in the second group. ∗ denotes the reproduced result.

| Method | Arch. | Road | Sidewalk | Building | Wall | Fence | Pole | Light | Sign | Veg | Terrain | Sky | Person | Rider | Car | Truck | Bus | Train | Motor | Bike | mIoU |
|---|---|---|---|---|---|---|---|---|---|---|---|---|---|---|---|---|---|---|---|---|---|
| ADVENT [85] | C | 72.9 | 14.3 | 40.5 | 16.6 | 21.2 | 9.3 | 17.4 | 21.2 | 63.8 | 23.8 | 18.3 | 32.6 | 19.5 | 69.5 | 36.2 | 34.5 | 46.2 | 26.9 | 36.1 | 32.7 |
| DANNet [89] | C | 84.3 | 54.2 | 77.6 | 38.0 | 30.0 | 18.9 | 41.6 | 35.2 | 71.3 | 39.4 | 86.6 | 48.7 | 29.2 | 76.2 | 41.6 | 43.0 | 58.6 | 32.6 | 43.9 | 50.0 |
| FREST [46] | T | 93.3 | 72.2 | 88.3 | 52.4 | 46.6 | 58.6 | 66.2 | 66.1 | 86.1 | 58.6 | 95.3 | 69.9 | 49.2 | 89.1 | 75.1 | 79.4 | 83.0 | 52.9 | 61.4 | 70.7 |
| DAFormer [38] | T | 58.4 | 51.3 | 84.0 | 42.7 | 35.1 | 50.7 | 30.0 | 57.0 | 74.8 | 52.8 | 51.3 | 58.3 | 32.6 | 82.7 | 58.3 | 54.9 | 82.4 | 44.1 | 50.7 | 55.4 |
| +DiDA | T | **68.5** | **52.5** | 82.5 | **43.4** | **42.7** | **59.8** | **60.3** | 53.4 | **75.7** | 40.2 | **63.7** | **59.3** | 30.4 | **87.3** | **68.6** | **76.1** | 79.5 | 41.9 | 36.6 | **59.1** |
| HRDA [39] | T | 88.3 | 57.9 | 88.1 | 55.2 | 36.7 | 56.3 | 62.9 | 65.3 | 74.2 | 57.7 | 85.9 | 68.8 | 45.7 | 88.5 | 76.4 | 82.4 | 87.7 | 52.7 | 60.4 | 68.0 |
| +DiDA | T | **90.7** | **65.4** | **89.3** | **58.3** | **50.1** | **68.7** | **70.8** | **66.6** | **79.1** | 46.2 | 67.1 | **73.0** | **49.4** | 85.2 | **85.9** | **89.4** | **91.5** | **56.3** | 60.1 | **70.7** |
| MIC* [40] | T | 90.1 | 65.0 | 87.7 | 55.5 | 43.3 | 60.6 | 63.8 | 66.2 | 75.8 | 54.3 | 85.0 | 69.5 | 47.4 | 88.6 | 80.7 | 89.5 | 88.8 | 55.4 | 59.1 | 69.8 |
| +DiDA | T | **90.5** | **68.6** | **89.0** | **62.9** | **50.1** | **65.4** | **66.3** | **68.8** | 75.6 | 51.9 | 84.3 | **70.0** | **51.4** | 88.1 | **82.4** | **92.5** | **92.2** | **56.6** | **63.6** | **72.1** |

# D  Additional Experiment Results

In this section, we evaluate the performance of our method on the CS.→ACDC adaptation task, which involves large domain shifts caused by adverse weather conditions such as rain, snow, and nighttime. As shown in Tab. 5, DiDA consistently improves the performance of multiple baseline models, including DAFormer, HRDA, and MIC, with a gain ranging from 2.3% to 3.7% mIoU. These improvements are highlighted in bold.

We attribute DiDA's superior performance in this scenario to the similarity between image degradation operations and real-world image corruptions caused by bad weather. Notably, the improvements are especially significant in classes that are typically affected by weather degradation, such as *traffic light*, *sign*, and *pole*. The degradation-aware training in DiDA (e.g., blur, noise, contrast reduction) helps the model learn more robust and generalizable features that align well with the characteristics of the ACDC dataset. In other words, DiDA implicitly narrows the domain gap by simulating weather-induced artifacts during training.

# E  DiDA Efficiency Analysis

In this section, we analyze the computational overhead introduced by DiDA in terms of GPU memory usage, iteration time, and total training time. As summarized in Table 6, while DiDA brings consistent

performance improvements across various UDA baselines, it also introduces moderate increases in computational cost.

Specifically, DiDA increases the GPU memory usage by approximately 60–70%, while it is computationally efficient, with an increase of 0.3-0.5s per iteration for each method. This overhead mainly stems from the additional modules introduced by DiDA, as illustrated in Algorithm 1. These include: (1) the diffusion-based degradation encoder $g'$, used to handle intermediate degraded domains; (2) the reconstruction head $h'$, which enforces semantic consistency via reconstruction and consistency losses ($\mathcal{L}^D$, $\mathcal{L}^R$); (3) additional forward passes on degraded inputs for both source and target domains. Despite this increase in resource usage, DiDA's design is modular and parallelizable, and the degradation-based learning can be seamlessly integrated with existing teacher–student frameworks. We argue that the trade-off is justified, as the performance boosts brought by DiDA (up to +3.7% mIoU) outweigh the moderate increase in training cost, especially in challenging real-world adaptation scenarios like CS.→ACDC.

To further enhance the efficiency of our method, we additionally provide a memory-friendly variant, denoted as DiDA ($g_{\text{time}}$). As shown in row 6 of Table 2, this version avoids the explicit use of the diffusion encoder, and instead implicitly models the $t$-specific semantic shift through a lightweight time embedding. Despite its simplified design, DiDA ($g_{\text{time}}$) still achieves a comparable performance gain when compared to the full version (row 6 vs. row 7). By eliminating the diffusion encoder, DiDA ($g_{\text{time}}$) significantly reduces GPU memory consumption—saving up to 15–20% compared to the full version—while also slightly accelerating the training process. This makes it a practical trade-off option for scenarios with limited hardware resources.

Furthermore, DiDA is only applied during the training phase. No architectural modifications or additional computations are introduced at inference time, ensuring that the inference speed and resource consumption remain identical to the baseline models.

Table 6: Computational resource requirements comparison

| Methods | GPU Memory (MB) | Time per iter (s) | Total Time (h) |
|---|---|---|---|
| DAFormer [38] | 9,807 | 1.32 | 14.5 (40K iters) |
| **+DiDA** ($g_{time}$) | 12754 | 1.59 | 17.8 (40K iters) |
| **+DiDA** | 15856 | 1.64 | 18.5 (40K iters) |
| HRDA [39] | 22325 | 2.11 | 23.5 (40K iters) |
| **+DiDA** ($g_{time}$) | 16727×2 | 2.58 | 28.7 (40K iters) |
| **+DiDA** | 19337×2 | 2.64 | 29.4 (40K iters) |
| MIC [40] | 22370 | 2.60 | 28.9 (40K iters) |
| **+DiDA** ($g_{time}$) | 16813 ×2 | 3.03 | 33.7 (40K iters) |
| **+DiDA** | 19343 ×2 | 3.13 | 34.8 (40K iters) |

## F    Influence of Parameter Settings

In this section, we further study the influence of parameter settings introduced in DiDA, i.e., DIC loss weight $\lambda_D$, reconstruction loss weight $\lambda_R$, and diffusion steps $T$. All experiments are conducted with DAFormer [38] on GTA→CS.

**DIC Loss Weight** $\lambda_D$**.** We first study the weight of the DIC loss $\lambda_D$ in Tab. 7. The weight for the DIC loss is sensitive to the UDA performance. Reducing the weight progressively diminishes performance until no DIC is used. A larger weight also results in decreased performance. If the weight is too large, such as $\lambda_D = 5$, it can lead to a significant decline in performance due to excessive disturbance.

**Reconstruction Loss Weight** $\lambda_R$**.** The weight of the reconstruction loss is also studied in a similar way (see Tab. 8). Based on the analysis above, we can draw a similar conclusion, with the difference being that this loss item has a slightly diminished impact compared to the previous one. The results demonstrate that values ranging from 1 to 10 consistently yield good UDA performance, providing a reasonably wide range for selecting robust hyperparameters.

**Diffusion Steps** $T$**.** We also study the value selection of diffusion steps $T$ (see Tab. 9). Notice that too large a value, like $T = 1000$, does not result in optimal performance, which is the default setting for DDPM [37] to facilitate high-quality generation. In the UDA setting, the number of iterations

and batch size is far less than the requirement to train the generative model. Therefore, we make adjustments to reduce the diffusion steps $T$, ensuring they are more suitable for this task. In our experiments, a value around 100 can achieve consistently good performance.

Table 7: Parameter study of loss weight $\lambda_D$.

| Weight $\lambda_D$ | 0 | 0.1 | 0.25 | 0.5 | 1 | 5 |
|---|---|---|---|---|---|---|
| mIou | 67.9 | 69.6 | 70.1 | **70.3** | 69.8 | 64.1 |

Table 8: Parameter study of loss weight $\lambda_R$.

| Weight $\lambda_R$ | 0 | 1 | 2.5 | 5 | 10 | 50 |
|---|---|---|---|---|---|---|
| mIou | 69.5 | 70.0 | 70.0 | **70.3** | 70.2 | 67.7 |

Table 9: Parameter study of diffusion steps $T$.

| Diffusion Steps $T$ | 10 | 50 | 100 | 200 | 1000 |
|---|---|---|---|---|---|
| mIou | 69.3 | 70.2 | **70.3** | 70.0 | 70.1 |

Table 10: Different strategies of time schedule.

| Time Schedule | linear | cosine | sigmoid |
|---|---|---|---|
| mIou | 69.8 | 70.2 | **70.3** |

Table 11: Results on GTA.→CS built with extended versions.

| Method | base | **B** | **M** |
|---|---|---|---|
| DAFormer [38] | 68.3 | 70.0 | 69.8 |
| HRDA [39] | 73.8 | 75.3 | 74.9 |
| MIC [40] | 75.9 | 76.7 | 76.6 |

Table 12: Results on SYN.→CS built with extended versions.

| Method | base | **B** | **M** |
|---|---|---|---|
| DAFormer [38] | 60.9 | 63.1 | 62.6 |
| HRDA [39] | 65.8 | 68.0 | 67.7 |
| MIC [40] | 67.3 | 68.7 | 68.2 |

## G  Influence of Time Schedule

We study different choices of time schedules for $\beta_t$, namely, linear [37], cosine [65], and sigmoid [44], with the same setting as Sec. F. The results are summarized in Tab. 10. Although selecting the appropriate schedule is significant for generating high-quality images in the diffusion model, it is robust in DiDA, and we can achieve consistently good performance among these strategies.

## H  Degradation-Based Domain Bridging

To further investigate the feasibility of our proposed motivation and provide deeper insight into the domain bridging mechanism, we conduct a quantitative analysis of the distribution discrepancy between domains. Specifically, we train models independently within different distribution spaces and evaluate their performance in terms of mIoU under both fully supervised and UDA settings, as shown in Fig. 7(a).

To better assess the degree of domain adaptation, we further report the relative performance in Fig. 7(b), defined as the ratio of UDA mIoU to the fully supervised counterpart at each degradation level. As the level of image degradation increases, we observe a gradual improvement in the relative performance, suggesting that the adaptation capability is enhanced. This indicates that domain-shared information is better preserved in the intermediate domains constructed via degradation.

However, image degradation inevitably destroys fine-grained visual details, which may include essential semantic cues. As a result, the semantic shift problem arises, where the corrupted features impair the discriminative power of the model and hinder further performance gains in UDA.

Our proposed DiDA framework addresses this issue by explicitly disentangling and compensating for the semantic shift through a diffusion-based encoder and reconstruction loss. This design enables the network to retain domain-invariant representations while mitigating the adverse effects of degradation, thereby achieving consistently better adaptation performance.

## I  More Results with Extended Versions

We further conduct more experiments about **Blur** (**B**) and **Mask** (**M**) built with DAFormer [38] and HRDA [39] and evaluate the performance on two benchmarks as shown in Tab. 11 and Tab. 12. These extended versions show consistent gains beyond all baselines, which further demonstrates the generality and expansibility of our framework.

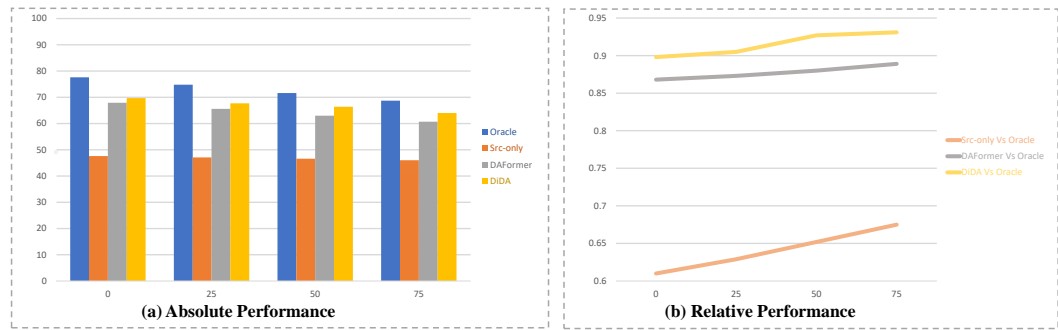

Figure 7: (a) The absolute performance of fully supervised learning and UDA settings (source-only, DAFormer [38] and DiDA) when the proportion of noise increases. Note that we train the models separately within different intermediate domains for the first three methods, while DiDA is trained only once and tested within different distributions. (b) Compared to oracle, the relative performance of UDA settings gradually improves, which means that the model's adaptability is strengthened in these intermediate domains.

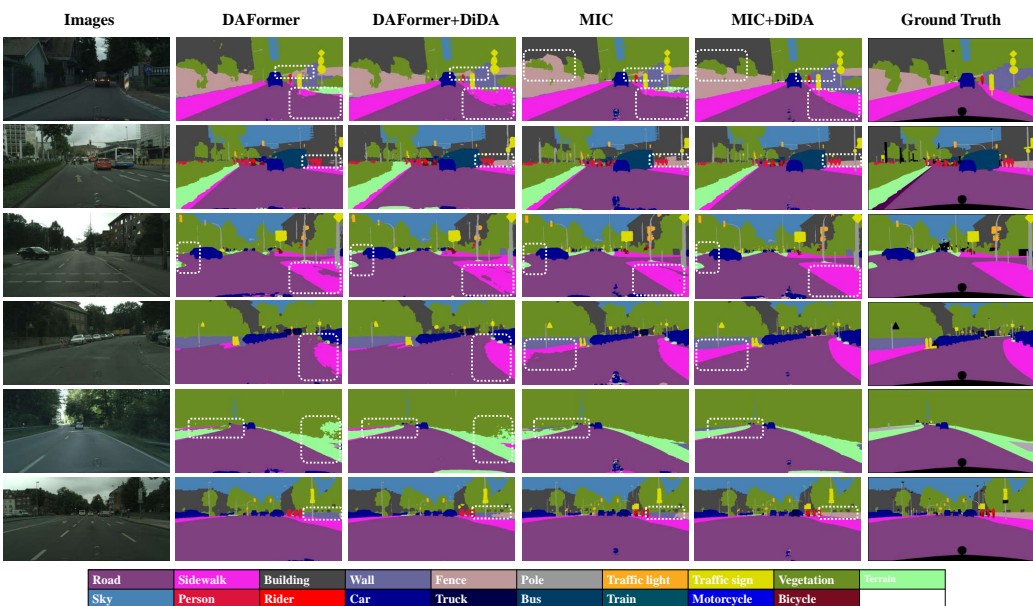

Figure 8: Qualitative comparison built with DAFormer and MIC. The dotted boxes mark regions improved by DiDA.

## J    Qualitative Results

In Fig. 8, we illustrate the qualitative comparisons of our DiDA against DAFormer and MIC. The previous methods fail to identify some classes on the target domain when their visual textures are significantly different from the source domain and confuse them with other visually similar classes (e.g., *sidewalk* and *road*, *fence* and *building*, *terrain* and *vegetation*). In this case, DiDA makes the model recognize semantic categories more dependent on context information, resulting in improved cross-domain performance.

## K    Visualization of Reconstruction Results

We visualize examples of reconstruction results in Fig. 9 with $t = 25, 50, 75$ in the time embedding module. The model can yield the best reconstruction results for degraded images with the corresponding $t$ value. A fixed value of $t$ leads to inadequate denoising for higher levels of degradation

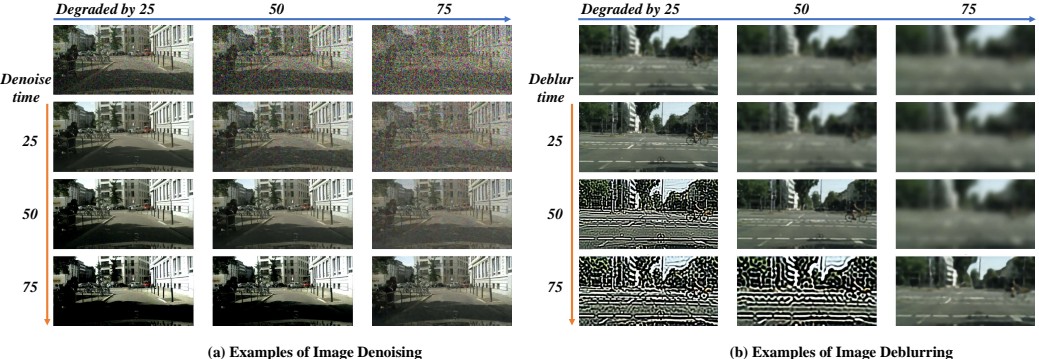

(a) Examples of Image Denoising        (b) Examples of Image Deblurring

Figure 9: The examples of image reconstruction results with fixed diffusion time $t = 25, 50, 75$ as input in the time embedding module, where the image is degraded by 25, 50, and 75, respectively.

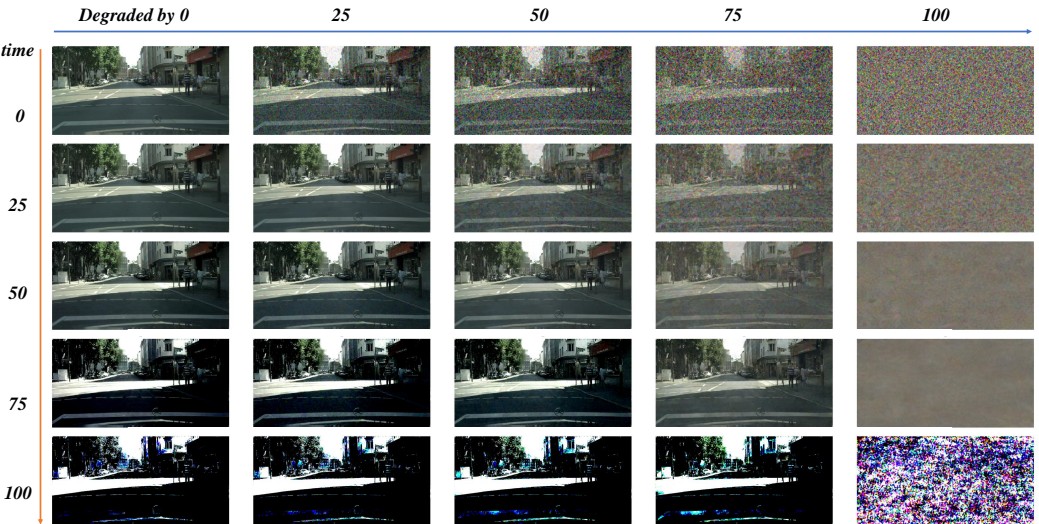

Figure 10: Examples of image denoising, horizontal coordinate represents the level of image degradation, and vertical coordinate indicates the $t$ used in the inference.

and excessive denoising for lower levels of degradation, meaning that degradation information is accurately encoded in the time embedding. This further alleviates the semantic shift phenomenon and facilitates adaptive learning from the disturbed image.

## L    More Examples for Image Reconstruction

In this section, we give further details, analysis, and examples of image reconstruction. For different versions of the implementation of DiDA, we execute the inference with different diffusion times $t$ on varying levels of image degradation for qualitative analysis.

**Image Denoising** This is the default setting of DiDA, the noise $\epsilon'$ is predicted by network $\bar{f}_\theta$ firstly and used to reconstruct the image as:

$$x_0^R = \frac{1}{\sqrt{\bar{\alpha}_t}}(x_t - \sqrt{1 - \bar{\alpha}_t}\epsilon_i'). \tag{14}$$

The examples are shown in Fig. 10. The noised images can be restored most perfectly when the diffusion time $t$ matches the level of degradation. We can observe that the higher value of $t$ has a more powerful anti-noise ability, which is consistent with our experimental results previously stated.

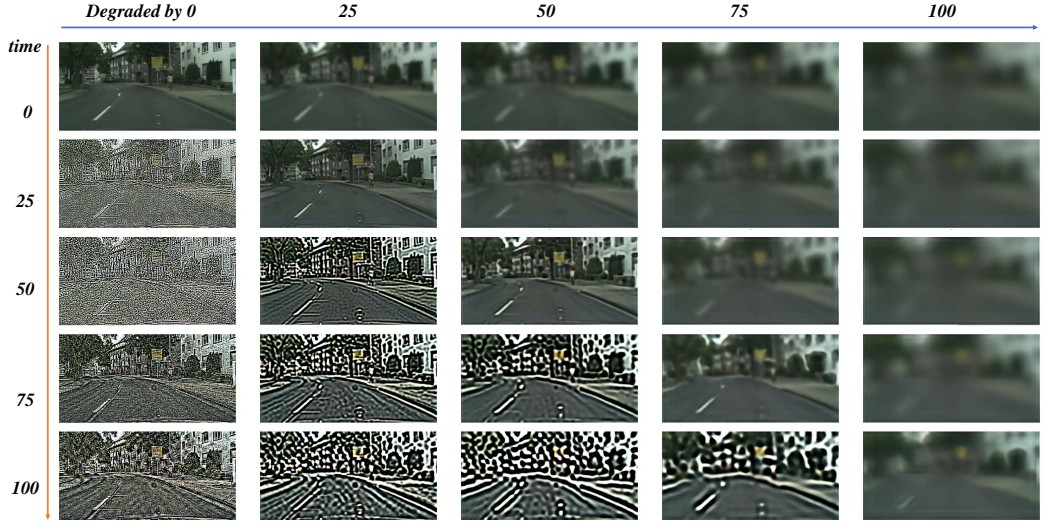

Figure 11: Examples of image deblurring, horizontal coordinate represents the level of image degradation, and vertical coordinate indicates the $t$ used in the inference.

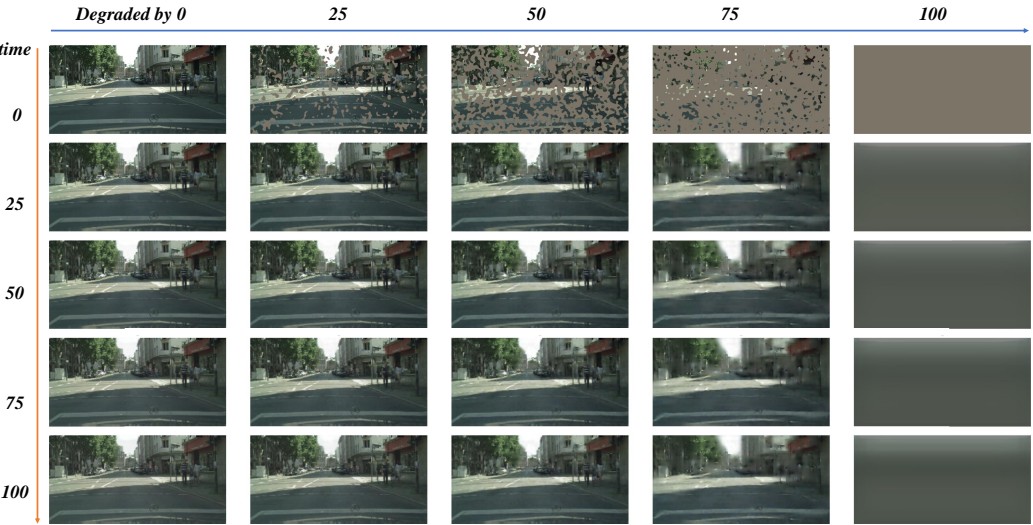

Figure 12: Examples of image inpainting, horizontal coordinate represents the level of image degradation, and vertical coordinate indicates the $t$ used in the inference.

**Image Deblurring.** In the following two settings, the reconstructed image is directly predicted by $\bar{f}_\theta$. Fig. 11 shows examples of image deblurring. We can draw a similar conclusion that the matched $t$ achieves the best performance as above. The higher $t$ results in excessive deblurring operation while the structure and edge information of the image are preserved, leading to more robust performance.

**Image Inpainting.** Fig. 12 shows examples of image inpainting. Unlike the former, when the variable $t$ changes in the inference, the reconstruction quality for the masked image does not vary significantly. Since the mask operation is operated in a global view to control the ratio while the previous degrading operation is executed locally, it is difficult for the network to sense the level of degradation in this case. Therefore, DiDA achieves slighter performance gains with the implementation of mask operation.

## M   More Discussion with Domain Bridging

Directly transferring knowledge from the source domain to the target domain can be challenging due to significant discrepancies and pixel-wise gaps between the domains. To address this issue, some works propose gradually transferring knowledge by building a bridge between the source and target domains. This is achieved by constructing intermediate domains at the image level [64, 92, 94], feature level [22, 23, 58], or output level [41, 98]. One line of work utilizes style transfer techniques [16, 19, 30] to transfer the style of source data to target data, effectively creating intermediate domains. Another approach leverages data mix augmentation techniques [82, 102, 7], such as CutMix [96] and Mixup [97], to construct various intermediate domains. While these methods can effectively reduce the domain gap and facilitate the adapting ability of models, they have certain limitations. Style transfer-based methods are often dataset-specific and may generate unexpected artifacts, while data mix strategies can disrupt the contextual distribution of images and require intricate designs. Both approaches lack generalization ability and face the semantic shift problem, where the intermediate domains may not preserve the semantic information of the original domains. These limitations restrict their applicability in different scenarios. In contrast, our work aims to explore a universal and concise domain bridging strategy that can be easily integrated into existing UDA frameworks while explicitly compensating for the discriminative representations. By constructing intermediate domains through a diffusion forward process, we propose a dataset-agnostic approach that alleviates the semantic shift problem and enhances the generalization ability of the model.

## N   Limitation and Societal Impact

Our work presents a general and modular approach for domain adaptive semantic segmentation. While our method demonstrates consistent improvements across multiple UDA baselines and datasets, it still presents a few limitations. First, DiDA introduces moderate computational overhead during training due to additional modules such as the degradation encoder and reconstruction losses. Although we provide a memory-friendly variant, further optimization is needed for extremely resource-constrained environments. Second, the choice and combination of degradation operations are currently heuristic and manually designed; automating or learning this selection process could further enhance performance and generality.

This work focuses on domain adaptive semantic segmentation, a key area in computer vision with broad applicability in domains such as autonomous driving, medical imaging, and remote sensing. At present, we are not aware of any direct negative societal impacts associated with the proposed method.

