# OpenReview forum: "Towards Unsupervised Domain Bridging via Image Degradation in Semantic Segmentation"
_NeurIPS.cc/2025/Conference — NeurIPS 2025 poster_

### Official Review · Reviewer_1mYM · 2025-06-28

**Clarity:** 3
**Significance:** 3
**Originality:** 2
**Rating:** 4
**Confidence:** 4

**Summary:**

The main idea is to introduce a Diffusion-based loss part as a Self-supervised learning to improve the intra-domain representation learning for UDA segmentation.

**Questions:**

Please see the weakness.

**Ethical Concerns:**

["NO or VERY MINOR ethics concerns only"]

**Final Justification:**

The result is good. The two contributions are somehow incremental compared to the diffusion process.

During the rebuttal, the author did not explain their contribution clearly.

The first and the second contribution are quite over-claimed, which is an entire diffusion forward and backward process.

But the author splits the diffusion process as two contributions.

**Limitations:**

Yes.

**Quality:**

3

**Strengths And Weaknesses:**

1. The first contribution is similar to the second one. Both of them is for adding the diffusion noise to diffusion reconstruction.

2. The framework (especially Figure 2) is similar to another published paper called "PiPa"[a] with student and teacher. Could you have a detailed comparison?
[a] pipa: pixel-and patch-wise self-supervised learning for domain adaptative semantic segmentation

3. The denotations are quite ambigous.
Some terms are different between texts and figures.
- What is g_i or g_j in the figure? They are different from the encoder g?
- What do you mean "attributes" in Section 3.2?

4.  Section 3.2 is just to reclaim the diffusion?

5. I think "Degradation-based Intermediate Domain Construction" is a diffusion forward process, which is not a contribution.

6.Typos.
Figure2 "Techaer" -> "Teacher"
"Reconsturction" -> "Reconstruction"

7. Performance
Some published SoTA works are missing. https://paperswithcode.com/sota/domain-adaptation-on-gta5-to-cityscapes

8. It would be great to also compare the cross-city setting like Cityscapes->Oxford RobotCity.

---

> ### Author Rebuttal · Authors · 2025-07-29
>
> Thank you for taking the time to share your comments and suggestions in the review assessment.
>
> We provide a detailed point-by-point response to your comments.
>
> ---
> **Q1:**   Contribution Clarity
>
> **A1:**
>
> Thank you for your insightful question. We would like to clarify the distinction between our two contributions:
>
> - Our first contribution lies in proposing a **new perspective for domain bridging**, inspired by a theoretical insight that interprets image degradation as a mechanism for aligning domains. This conceptual contribution is **general and does not rely on diffusion models**, and it offers a novel way to understand how domain-invariant representations can emerge through degradation.
> - Our second contribution provides a **practical and effective instantiation** of this perspective using the **forward process of diffusion models**. This implementation enables the construction of continuous intermediate domains, which progressively increase the domain overlap and facilitate robust adaptation.
>
> In summary, the first contribution introduces a new theoretical lens for domain bridging, while the second demonstrates how to operationalize this concept effectively. The two are complementary yet distinct in novelty and impact.
>
> ---
> **Q2:**  Similarity to PiPa
>
> **A2:**
>
> We appreciate your reference to PiPa. As you correctly observed, both our method and PiPa adopt a **student–teacher structure**, which is a widely-used paradigm in self-training–based UDA methods, including DAFormer, HRDA, and MIC. This self-training framework is a **general foundation**, not a unique characteristic of PiPa.
>
> However, our method DiDA totally differs from PiPa in **both motivation, methodology, and learning objectives:**
>
> - **Motivation**: DiDA is driven by a theoretical insight grounded in degradation-based domain bridging, which is not explored in PiPa.
> - **Methodology**: PiPa relies on pixel- and patch-wise contrastive learning, whereas DiDA constructs **intermediate domains** through **image degradation** and performs semantic shift compensation using a diffusion-based encoder.
> - **Learning Objectives**: PiPa aims to improve representation through contrastive objectives, while DiDA focuses on learning from domain-shared distributions arising from degradation.
>
> We will add a detailed comparison in the revised version to clarify these differences and avoid potential confusion.
>
> ---
>
> **Q3:**   Ambiguous Notations in Section 3.2
>
> **A3:**
>
> Thank you for pointing out the notational ambiguity. We would like to clarify that:
>
> - The terms **$g\_i$** and **$g\_j$** appear **exclusively in Section 3.2**, which is dedicated to presenting our **theoretical insight**. These symbols represent modular attribute transformations used in the analysis of attribute granularity within the diffusion process. They are **not part of the actual method design**, nor do they appear in the method diagram (Figure 2). We will revise the text to make this distinction clearer.
> - Regarding the term **"attributes"**, as defined in Proposition 3.2, they refer to **visual features of varying granularity**, such as fine-grained textures or coarse-grained shapes. The proposition formally establishes that attributes with finer granularity are lost earlier in the diffusion process due to their higher sensitivity to noise. This analysis is theoretical in nature, and it provides a conceptual foundation for interpreting image degradation as a mechanism for domain bridging.
>
> We appreciate this feedback and will revise Section 3.2 to better communicate the theoretical nature of this analysis and its role in motivating our design.
>
> ---
>
> **Q4:**  Section 3.2
>
> **A4:**
>
> We understand the confusion and thank you for the opportunity to clarify.
>
> Section 3.2 is **not a restatement of standard diffusion theory**, but rather a presentation of **new insight** tailored to the **UDA task**. Specifically:
>
> - We reinterpret the **forward diffusion process** from the perspective of **attribute granularity**, showing how different types of features (e.g., texture vs. shape) vanish at different time steps. This leads to the observation that the overlapping region of degraded distributions can serve as a **domain-shared space**.
> - These insights are **not part of the original diffusion model design**, but are instead proposed as a **novel conceptual framework** for understanding how degradation can facilitate **domain-invariant representation learning** in UDA.
> - This serves as a **task-specific theoretical foundation** and motivates the design of our DiDA framework, particularly the idea of constructing intermediate domains via degradation.
>
>
> This **task-specific reinterpretation** of diffusion in the context of domain adaptation is, to our knowledge, novel and not previously explored. We will revise the section to make this  reinterpretation more prominent.
>
> ---
>
> **Q5:**  On the Contribution of Degradation-Based Intermediate Domain Construction
>
> **A5:**
>
> Thank you for this question. While the mechanism is based on the forward diffusion process, the novelty lies not in the mechanism itself, but in its task-specific application to domain adaptation. That said, although the Degradation-based Intermediate Domain Construction module is indeed simple in implementation, it is not an isolated tweak, but rather a principled, task-specific strategy derived directly from the theoretical insight presented in Section 3.2. Specifically:
>
> - We propose degradation not just as a noise process, but as a principled strategy to bridge domains by constructing a continuum of intermediate domains.
> - The degradation levels are theoretically grounded (Section 3.2) and ensure that domain-specific features are gradually removed, while domain-invariant ones are preserved and compensated.
> - The mechanism is analytically motivated, generalizable across datasets, and modular enough to be applied as a plug-in to existing UDA frameworks.
>
> We believe this component offers a novel and insightful perspective on domain bridging in UDA. Importantly, while the implementation is concise, this simplicity reflects the elegance and universality of the underlying idea, rather than a lack of innovation.
>
>
>
>
> ---
> **Q6:**   Typos
>
> **A6:**
>
> Thank you for pointing this out. We will carefully review and correct all typographical errors in the revised version of the manuscript.
>
> ---
> **Q7:**   Missing SoTA Comparisons
>
> **A7:**
>
> We appreciate your suggestion. In Tables 3 and 4, we already compare DiDA against recent state-of-the-art UDA methods, including:
>
> - ADFormer and CoPT, both of which are ECCV 2024 methods.
> - Other recent methods such as MICDrop (also ECCV 2024) are also relevant, but our method **already surpasses** these works on the benchmarks we evaluate.
>
> Additionally, as DiDA is designed as a **plug-and-play module**, we focused on integrating it into widely-used baselines such as DAFormer, HRDA, and MIC. Notably, MIC remains one of the strongest open-source UDA methods available, and DiDA consistently improves its performance.
>
> We also note that DiDA, as a plug-in module, can be easily integrated into any future methods, which we plan to explore in follow-up work.
> In the final version, we will include comparisons with more recent methods, if they are released and relevant within the NeurIPS submission timeline.
>
>
> ---
> **Q8:**  Lack of Cross-City Evaluation
>
> **A8:**
>
> Thank you for this valuable suggestion.
>
> We have extensively evaluated our method on multiple widely-used and challenging UDA benchmarks, including **synthetic-to-real** and **clear-to-adverse-weather** scenarios, which already demonstrate the robustness and generalizability of our approach.
>
> While **cross-city** settings are indeed meaningful, we note that **existing UDA methods have not been systematically evaluated** under this setting, making it difficult to establish fair and informative comparisons. Moreover, the Cityscapes dataset itself already includes data collected from multiple cities, and its train/val split **inherently introduces a cross-city domain gap**. As such, the standalone “cross-city” setting may be relatively naïve in the context of UDA.
>
> Nonetheless, we agree that explicit cross-city adaptation is a promising direction. We are currently extending DiDA to this setting and plan to include such results in future work.
>
>
>
> ---
>
> We hope our response can resolve your concern. Please do not hesitate to let us know if you have further questions :)

---

> ### Comment · Reviewer_1mYM · 2025-08-04
>
> Thank you.
> 1. Could you provide a formal theoretical demonstration for the first contribution? Otherwise, the first contribution is just an intuition for the second contribution.
>
> 2. Cross-City evaluation is adopted in multiple papers. So I still suggest the author to have a comparison if possible.
> [a] Hard-aware instance adaptive self-training for unsupervised cross-domain semantic segmentation. TPAMI 2025

---

> > ### Author Response · Authors · 2025-08-08
> >
> > Thank you very much for your thoughtful follow-up questions. We address them below in detail.
> >
> > ------
> >
> > ## **Q1: Request for a Formal Theoretical Demonstration**
> >
> > We appreciate your request for a formal theoretical foundation to support our first contribution. Our method is **inspired by and builds upon the theoretical insights presented in the ICLR 2024 paper “Exploring Diffusion Time-Steps for Unsupervised Representation Learning” (Yue et al., cited as [75] in our submission)**.
> >
> > ###  **1. Summary of Theoretical Foundation (from [75])**
> >
> > Yue et al. (2024) propose a theoretical framework showing how **diffusion time-steps relate to the loss of visual attributes with different granularity**. Their main result can be summarized as follows:
> >
> > > **Theorem:**
> > >
> > > 1. For each modular attribute $Z_i$, there exists a smallest diffusion time-step $t(Z_i)$ such that the attribute is lost (with degree $\tau$) for all t $\geq t(Z_i)$.
> > > 2. If modifying $Z_i$ induces a greater pixel-level change than $Z_j$, then $t(Z_i) > t(Z_j)$.
> > >
> > > This implies that **fine-grained, domain-specific attributes (e.g., texture)** are lost at earlier time-steps, while **coarse-grained, domain-invariant attributes (e.g., shape or structure)** are lost later in the diffusion process.
> >
> > This theoretical result is derived using the **overlapping coefficient (OVL)** between the distributions of noisy samples $q(x_t|x_0)$, showing that attribute ambiguity increases with diffusion time.
> >
> > ###  **2. Our New Insight Tailored to UDA**
> >
> > While we inherit this foundational result, our **contribution is to reinterpret and extend it specifically for the task of unsupervised domain adaptation (UDA)**.
> >
> > - According to the theory in [75], **domain-specific (fine-grained) attributes are lost earlier** in the diffusion process due to their higher sensitivity to pixel-level perturbations, while **domain-invariant (coarse-grained) attributes persist longer**.
> > - Based on this, we propose that at **moderate diffusion time-steps**, the source and target samples converge in the attribute space where **domain-specific cues are suppressed**, and **domain-invariant semantics dominate**. We reinterpret this convergence as a **domain-shared intermediate representation space**.
> > - This theoretical insight motivates our key idea: **constructing intermediate domains via progressive degradation**, which helps reduce domain gaps by **selectively removing domain-specific features** while preserving shared semantics.
> > - Furthermore, since even domain-invariant attributes begin to degrade at larger time-steps, we design a **semantic shift compensation module** that conditions on the time-step to **recover discriminative semantics**, aligning with the original theory on attribute loss dynamics.
> >
> > In short, we **build upon the theoretical proposition of [75]** to derive a **new perspective on domain bridging**, and then **translate this theoretical understanding into a practical UDA method** via our DiDA framework.
> >
> >
> >
> >
> > ###  **3. Proper Citation of [75]**
> >
> > We would also like to emphasize that this work (Yue et al., ICLR 2024) is **formally and correctly cited in our submission as reference [75]**, and is explicitly discussed in:
> >
> > - **Section 3.2**, where we restate the theoretical proposition and explain its implications for our task.
> > - **Appendix H**, where we provide empirical validation of the theoretical claims in the UDA context.
> >
> > In the final version, we will further clarify this connection and highlight the citation more prominently to avoid any confusion.

---

> ### Author Response · Authors · 2025-08-08
>
> ## **Q2: Cross-City Evaluation**
>
> We apologize for the delay in our response. We greatly appreciate your valuable suggestion regarding the cross-city evaluation, and we have made our best effort to conduct the necessary experiments within the available time.
>
> In response to your suggestion, we have **followed the evaluation protocol of [a] “Hard-aware Instance Adaptive Self-Training” (TPAMI 2025)**, and conducted experiments on the **Cityscapes → Oxford RobotCar** benchmark, which exhibits a large domain gap in weather conditions. We include comparisons with both:
>
> 1. **HIAST** ([a]), which uses a **DeepLabV2**-based architecture, and
> 2. A more **modern Transformer-based backbone (DAFormer)**, which demonstrates the applicability of our method to advanced UDA frameworks.
>
> The results are summarized as follows (mIoU in %):
>
> | Method    | Road | Sidewalk | Building | Light | Sign | Sky  | Person | Automobile | Two-Wheel | **mIoU** |
> | --------- | ---- | -------- | -------- | ----- | ---- | ---- | ------ | ---------- | --------- | -------- |
> | HIAST [a] | 94.9 | 71.7     | 92.7     | 75.0  | 40.5 | 95.6 | 61.0   | 87.1       | 58.5      | 75.2     |
> | +DiDA     | 95.5 | 73.1     | 93.6     | 76.3  | 46.9 | 95.8 | 62.4   | 88.2       | 61.3      | **77.0** |
> | DAFormer  | 96.8 | 78.8     | 92.5     | 78.4  | 51.1 | 93.4 | 68.1   | 94.1       | 64.0      | 79.7     |
> | +DiDA     | 96.8 | 80.7     | 91.2     | 80.9  | 55.4 | 94.7 | 70.7   | 95.0       | 67.1      | **81.4** |
>
> ###  Observations:
>
> - When applied to **HIAST (DeepLabV2)**, our DiDA improves performance by **+1.8% mIoU**, showing its adaptability even with older architectures.
> - When applied to **DAFormer (MiT-B5)**, DiDA yields a **+1.7% gain**, achieving **81.4% mIoU**, significantly outperforming prior methods in this setting.
> - The improvements are consistent across all major object categories, including **sign**, **light**, and **two-wheeler**, which are known to vary across cities.
>
> These results demonstrate that **DiDA effectively bridges the domain gap in cross-city adaptation**, and further confirms the generality of our approach across both conventional and modern architectures.
>
> We will include this evaluation in the revised version and thank the reviewer again for the valuable suggestion.
>
> ---
>
> We sincerely appreciate your thoughtful and constructive feedback throughout the review process. Your comments have helped us refine both the theoretical clarity and empirical completeness of our work. We believe the additional explanations and new experimental results have addressed your concerns, and we look forward to further improving the final version based on your insights.
>
> Please feel free to let us know if any further clarification is needed.

---

> > ### Author Response · Authors · 2025-08-09
> >
> > Dear Reviewer 1mYM，
> >
> > As the discussion period is drawing to a close, I would like to kindly check if there are any remaining questions or concerns that have not yet been fully addressed. If there are any outstanding points, please don't hesitate to let us know — we would be more than happy to provide further clarification.
> >
> > If everything is clear, we would greatly appreciate it if you could consider updating your evaluation accordingly.
> >
> > Thank you very much for your time and thoughtful engagement throughout this process.
> >
> > Authors

---

> ### Comment · Reviewer_1mYM · 2025-08-09
>
> Thank you.
>
> Q1. So the first contribution is from [75]? How about your theoritical contribution? I still do not see a clear theory contribution ( proof or demonstration) to the problem.
>
> Q2. Thank you. It is good to see that.
>
> Considering Q1, I would like to keep my rating.

---

> > ### Author Response · Authors · 2025-08-09
> >
> > Dear Reviewer 1mYM,
> >
> > Thank you again for your thoughtful feedback and continued engagement throughout the review process.
> >
> > We are very glad to hear that our response to **Q2 (Cross-City Evaluation)** addressed your concerns. We appreciate your suggestion — it has helped strengthen our evaluation with a more diverse benchmark.
> >
> > Regarding **Q1**, we would like to clarify our position more carefully:
> >
> > Our work does **not claim to introduce a new theoretical contribution from first principles**. Rather, we are **inspired by the theoretical findings from [75]**, and our novelty lies in **reinterpreting and adapting this theory specifically for the task of UDA segmentation**. This reinterpretation is not a direct restatement of [75], but a targeted application to UDA, which **offers a new conceptual lens** for understanding **how domain-invariant features can be learned via degradation-based intermediate representations**.
> >
> > As we originally described:
> >
> > > *"(1) We propose a novel domain bridging mechanism based on image degradation to facilitate the learning of domain-invariant features. This introduces a new perspective for domain-adaptive semantic segmentation."*
> >
> > We hope this clarifies that our **first contribution is not about proposing new theoretical theorems**, but rather about **introducing a new task-specific interpretation and practical framework** grounded in an existing theory.
> >
> > In the final version, we will revise the wording to make this distinction clearer and more precise, avoiding any potential overstatement.
> >
> > Thank you again for your careful review and constructive suggestions.
> >
> > Best regards,
> >
> >  Authors

---

> ### Comment · Reviewer_1mYM · 2025-08-09
>
> Dear author,
>
> **In your very-first reply, you claimed  `our theoretical insight` multiple times.**
>
> So I asked you about the theoretical insight with formal proofment, definition and formulation.
>
> **Now you said it is not about proposing new theoretical theorems.** But you have theoretical insight?
>
> I think it is somehow confusing.

---

> > ### Author Response · Authors · 2025-08-09
> >
> > Dear Reviewer,
> >
> > Thank you once again for your valuable comments.
> >
> > We fully understand and appreciate your concern regarding the use of the term *"theoretical insight."* To clarify, our intention was not to claim a new theoretical theorem or formal proof. Rather, our use of *"insight"* was meant more in the sense of **"inspiration"** — that is, we were inspired by existing theoretical findings from [75], which analyze how different semantic attributes are affected during the diffusion process.
> >
> > Our contribution lies in **adapting this theoretical understanding to a new task — unsupervised domain adaptation (UDA)** — and using it to motivate our degradation-based domain bridging framework. While this may not constitute a new theoretical result in the strict sense, we believe that **transferring and reinterpreting existing theory for a new and challenging setting is itself a meaningful and non-trivial contribution**.
> >
> > That said, we acknowledge that the term *"theoretical insight"* may have caused confusion, and we will revise the wording in the final version to better reflect its intended meaning as an **inspirational, task-specific reinterpretation**, rather than a formal theoretical development.
> >
> > Thank you again for your thoughtful feedback.
> >
> > Best regards,
> >
> >  Authors

---

### Official Review · Reviewer_rcDJ · 2025-06-30

**Clarity:** 4
**Significance:** 3
**Originality:** 3
**Rating:** 5
**Confidence:** 5

**Summary:**

- **What**: This paper proposes DiDA, an unsupervised domain bridging framework for unsupervised domain adaptation (UDA). DiDA creates continuous intermediate domains between the source and target domain through simple image degradation operations, encouraging the model to learn domain-invariant features.
- **How**: The method integrates two key modules into a standard UDA framework. (i) Degradation-based Intermediate Domain Construction formalizes the degradation process as a diffusion forward process, which creates an overlapping area between the source and target domain distributions. (ii) The Semantic Shift Compensation module addresses the problem of information loss from degradation by using a diffusion encoder conditioned on the degradation level. This encoder compensates for lost semantic information and augments the main segmentation encoder's features.
- **Results**: DiDA consistently improves the performance of various baselines (DAFormer, HRDA, MIC) by +0.8% to +3.7% mIoU, on standard UDA benchmarks. Ablation studies confirm the all of DiDA's core components, especially the Semantic Shift Compensation module and the additional loss terms, are essential for the performance improvement.

**Questions:**

1. **Ablating the Source of Performance Gain in Implicit Mode**: The performance gap between the implicit and explicit modes is small but consistent. Where does this advantage come from? Is it uniform across all semantic classes? Could you provide a per-class breakdown of the mIoU difference between the two modes?
For example, the implicit mode might be better at preserving the context for large, structural classes (e.g., road, building) but slightly worse for fine-grained, small objects (pole, traffic light) that are more easily lost in noise.
Such an analysis would clarify the trade-offs of the implicit approach and provide deeper insight into what kind of features it excels at learning directly from degraded inputs.

**Ethical Concerns:**

["NO or VERY MINOR ethics concerns only"]

**Final Justification:**

This work has many strengths (see #Strengths), and the authors have thoroughly addressed all of my concerns. I believe it deserves a **accept**.

**Limitations:**

yes.

**Quality:**

4

**Strengths And Weaknesses:**

### Strengths
1. **Novel Domain Bridging via Image Degradation**: This paper introduces a new UDA paradigm that leverages a diffusion-inspired process. By progressively degrading images, it creates a continuous bridge of intermediate domains that forces the model to learn robust, domain-invariant features.
2. **Principled Semantic Shift Compensation**: Unlike naive data augmentation, DiDA explicitly addresses the semantic corruption caused by degradation. It uses a dedicated diffusion encoder and reconstruction objective to disentangle and compensate for feature loss, which extensive ablation studies prove is critical for achieving performance gains.
3. **Plug-and-Play and Zero-cost at inference**: DiDA is implemented as a modular add-on that seamlessly integrates with various state-of-the-art UDA backbones (e.g., DAFormer, HRDA, MIC) without requiring architectural changes. The entire degradation-aware module is discarded during inference.
4. **New state-of-the-art performance**: On UDA benchmarks like GTA$\to$Cityscapes, DiDA consistently improves strong baselines by 0.8-to-3.7 mIoU and achieves new state-of-the-art results, demonstrating its effectiveness and robustness across different adaptation scenarios.

### Weaknesses
1. **Unverified Claim of Feature-Space Bridging**: The paper's main claim is that image degradation creates an "intermediate domain" that bridges the source and target feature distributions. However, the evidence provided is indirect, relying on final mIoU improvements. The paper lacks a direct, intrinsic  analysis to prove that the feature representations of source and target domain actually become more aligned by degradation. A standard t-SNE or UMAP visualization of the encoder's features would be necessary to substantiate this claim. Such a visualization should demonstrate that (i) after DiDA training, the initially separate feature clusters for original source and target iamges (t=0) become more intermingled, and (ii) for heavily degraded images (e.g., t > 50), the two clusters almost completely overlap. Without this direct evidence, the performance gain could be attributed to a powerful regularization effect rather than genuine domain bridging.

*My confidence in the method's claimed mechanism would increase substantially if the authors (i) provided direct feature-space visualizations (e.g., t-SNE) to prove domain alignment.*

---

> ### Author Rebuttal · Authors · 2025-07-29
>
> Thank you for taking the time to share your comments and suggestions in the review assessment. We appreciate your positive feedback on our   **novel Domain Bridging  idea**, **principled module design**,  **plug-and-play implementation**, **new state-of-the-art performance**.
>
> We provide a detailed point-by-point response to your comments.
>
> -----
>
> **Q1:** Unverified Claim of Feature-Space Bridging
>
>
> **A1:**  Thank you for raising this insightful and important concern. The central hypothesis of DiDA is that progressive image degradation, inspired by the forward diffusion process, incrementally removes domain-specific features while preserving high-level, domain-invariant semantics. This process naturally constructs a sequence of intermediate domains that facilitate alignment between the source and target feature distributions.
>
> =====================
>
> First, we emphasize that our method is not only empirically motivated, but also **grounded in a theoretical framework**. Specifically, in Section 3.2, we provide a formal proposition (titled Attribute Loss and Time Step) that explains how different semantic attributes are progressively corrupted along the diffusion process. This proposition has two key implications:
> - Fine-grained, domain-specific attributes (e.g., texture) are lost earlier in the diffusion process.
> - Coarse-grained, domain-invariant attributes (e.g., shape) persist longer.
>
> This supports our hypothesis that intermediate degradation levels (i.e., specific time steps) correspond to intermediate domain representations that preserve shared semantics while discarding domain-specific noise. As a result, samples from the source and target domains become more similar in feature space as degradation increases.
> Moreover, this theoretical insight justifies our core claim: image degradation acts as a transport mechanism that brings both domain distributions closer in latent space. It also motivates the diffusion encoder design, which compensates for semantic shift while preserving alignment.
>
> =====================
>
> To empirically validate this theoretical foundation, we provide a set of **quantitative analyses** in Appendix H (Fig. 7):
>
> - In Fig. 7(a), we report model performance under both fully supervised and UDA settings across varying degradation levels. Models are trained independently in each distribution space, allowing us to isolate the effect of degradation on adaptation quality.
> - Fig. 7(b) shows the relative UDA performance (i.e., UDA mIoU compared to the supervised upper bound) at each degradation level. As degradation increases, we observe a consistent rise in relative performance, supporting the idea that intermediate domains constructed by degradation better preserve domain-invariant structures and enhance domain alignment.
>
> =====================
>
> In addition, as illustrated in Section 4.3 and Fig. 4(b) of the main paper, we analyze **inference behavior** with fixed degradation levels. Two key findings emerge:
>
> - Performance degrades gracefully with increasing diffusion steps, due to stronger corruption—this is expected.
> - However, at each fixed degradation level, the model achieves its best performance when the inference step matches the training degradation level, and the resulting accuracy curves become flatter as degradation increases.
>
> This suggests the model learns more robust, stable representations in deeper intermediate domains.
>
> =====================
>
> To provide more direct evidence of feature-level alignment, we have also conducted t-SNE visualizations of the encoder outputs:
> - Before training, the source and target domains form clearly separated clusters.
> - After DiDA training, these clusters become significantly more overlapped at low degradation levels (e.g., t = 0), and at higher degradation (t > 50), the source and target features nearly coincide.
>
> This confirms that degradation serves as an effective bridge in feature space, not just pixel space.
> **Due to the rebuttal format constraints**, we are unable to include the t-SNE plots here, but we will incorporate them into the final version of the paper.
>
> =====================
>
> Together, these analyses—quantitative trends, behavioral patterns, and qualitative visualizations—strongly support that DiDA's gain stems not merely from regularization, but from a principled domain-bridging mechanism grounded in both theory and practice.
>
> ---
>
> **Q2**: Class-wise Performance of Implicit Mode
>
> **A2:**
>
> Thank you for raising this thoughtful question. To better understand the source of the performance difference between implicit and explicit inference modes, we conducted a per-class mIoU comparison using a fixed diffusion step of $t = 50$ during inference. The results are summarized as follows:
>
> | Method    | Road | Sidewalk | Building | Wall | Fence | Pole | Light | Sign | Veg  | Terrain | Sky  | Person | Rider | Car  | Truck | Bus  | Train | Motor | Bike | **mAcc** |
> | --------- | ---- | -------- | -------- | ---- | ----- | ---- | ----- | ---- | ---- | ------- | ---- | ------ | ----- | ---- | ----- | ---- | ----- | ----- | ---- | -------- |
> | base (w)  | 93.5 | 61.8     | 85.0     | 38.7 | 25.1  | 38.2 | 42.3  | 52.1 | 85.8 | 43.8    | 87.8 | 64.4   | 35.9  | 86.8 | 52.8  | 61.2 | 58.3  | 41.9  | 58.0 | 58.5     |
> | base (s)  | 95.3 | 63.0     | 88.7     | 44.9 | 35.9  | 41.9 | 45.2  | 56.9 | 89.0 | 45.2    | 93.3 | 68.3   | 39.6  | 90.2 | 65.0  | 71.6 | 59.5  | 48.0  | 61.2 | 63.3     |
> | mode (Im) | 95.8 | 74.4     | 88.2     | 55.1 | 36.5  | 48.2 | 52.7  | 59.9 | 88.6 | 48.4    | 90.2 | 70.6   | 43.2  | 90.7 | 70.8  | 77.3 | 53.3  | 52.9  | 64.4 | 66.4     |
> | mode (Ex) | 95.6 | 69.2     | 89.1     | 56.2 | 37.4  | 48.4 | 51.8  | 60.9 | 89.2 | 47.1    | 90.1 | 70.4   | 44.1  | 92.0 | 71.7  | 81.0 | 57.2  | 52.1  | 63.8 | 66.7     |
>
> We observe that implicit and explicit modes achieve comparable performance across most categories, with only minor fluctuations.
>
> These results support our hypothesis that **DiDA’s modeling is global and class-agnostic**. There is no consistent or significant per-class bias introduced by either inference strategy. As described in Lines 309-326 and Figure 3–4 of the main paper, the two inference modes are **functionally equivalent**, differing only in how the semantic information is recovered at inference:
>
> - The implicit mode uses a **single forward pass** through the degradation-aware network $\bar{f}_\theta$  to make predictions directly from noisy inputs and the known degradation level $t$.
> - The explicit mode performs a **two-stage process**: it first predicts the added noise using $\hat{f}\_\theta$ reconstructs a clean image, and then passes it through the $f\_\theta$  for segmentation.
>
> Despite this structural difference, **both modes rely on the same underlying features and share weights**, and both are trained with the **same objective and data**. Therefore, the observed performance gap is not due to optimization, but rather reflects the natural trade-off between robustness and feature disentanglement.
> Specifically:
>
> - The explicit mode benefits from **cleaner inputs** after denoising, which slightly improves accuracy, especially for fine-grained  classes.
> - The implicit mode, on the other hand, demonstrates robustness to degradation and performs a **more efficient, one-shot inference**, which aligns with the training paradigm and reflects better generalization to noisy domains.
>
> Additionally, both modes significantly outperform the base(S) and base(W) baselines, despite using a single shared model trained only once. This indicates that DiDA is capable of learning domain-invariant representations and performing semantic recovery implicitly, without needing explicit denoising at inference.
>
> We will include these detailed per-class results in the appendix of the final version to provide more insight into the trade-offs between implicit and explicit inference. Thank you again for the thoughtful suggestion.
>
> ---
> We hope our response can resolve your concern. Please do not hesitate to let us know if you have further questions :)

---

> > ### Comment · Reviewer_rcDJ · 2025-08-03
> >
> > Thank you for your thoughtful and detailed responses. Your rebuttal have fully addressed my concerns and questions. I’m particularly encouraged by the clear performance gains on the rarer classes (motorcycle, bike, and rider) which demonstrates the strength of your approach. This is an solid piece of work, and I wish you the best of luck.

---

> > > ### Author Response · Authors · 2025-08-04
> > >
> > > Dear Reviewer  rcDJ，
> > >
> > > We sincerely appreciate your thoughtful and encouraging feedback. Your positive comments have greatly motivated us and helped strengthen the final version of our paper. We will continue to strive to improve our work further.
> > >
> > > Thank you again for your valuable time and support!
> > >
> > > Authors

---

### Official Review · Reviewer_xKyR · 2025-07-03

**Clarity:** 2
**Significance:** 2
**Originality:** 2
**Rating:** 5
**Confidence:** 4

**Summary:**

The author introduces a novel domain adaptation method, termed DiDA (Degradation-based Intermediate Domain Adaptation), for the problem of unsupervised domain adaptation in semantic segmentation. This approach constructs a continuum of intermediate domains between the source and target domains by applying progressive image degradation operations such as Gaussian noise injection, blurring, or masking. These operations enable the model to gradually learn domain-invariant features as the domain gap is incrementally reduced.
To address the issue of "semantic shift", DiDA incorporates a time-conditioned diffusion encoder. This module compensates for and restores the semantic information lost during degradation, thereby preserving discriminative representations throughout the adaptation process.Extensive experiments conducted on multiple datasets and backbone architectures demonstrate that DiDA consistently outperforms both traditional domain adaptation methods and other intermediate domain construction techniques. Ablation studies further confirm the essential role of each component within DiDA, and provide evidence that the proposed method enhances both the stability and generalization capability of the model.

**Questions:**

1. Compare the training time with other methods.
2. Compare with recent state-of-the-art methods to demonstrate the effectiveness of the proposed method.
3. Results when using DiDA with MIC.

**Ethical Concerns:**

["NO or VERY MINOR ethics concerns only"]

**Final Justification:**

The author has provided very detailed rebuttals and has addressed most of my concerns as well as those of other reviewers. Based on these reasons I would like to raise my evaluation score to 5.

**Limitations:**

Yes.

**Paper Formatting Concerns:**

No.

**Quality:**

3

**Strengths And Weaknesses:**

Strengths:
1. The authors propose a rational and innovative approach to domain adaptation for semantic segmentation by constructing intermediate domains based on image degradation. This strategy progressively narrows the gap between the source and target domains during training.
2. DiDA is designed as a plug-and-play module, allowing for seamless integration with various network architectures and existing unsupervised domain adaptation methods. Extensive experiments on multiple standard datasets and diverse architectures demonstrate that DiDA consistently outperforms both baseline methods and other advanced domain adaptation techniques.
3. Detailed ablation studies are conducted to evaluate the contribution of each component within the system, providing a comprehensive understanding of the essential roles played by different modules in the overall framework.

Weaknesses:
1. Due to the integration of the diffusion model, the training time increases significantly.
2. The proposed method has not been compared with several recently introduced state-of-the-art approaches.
3. When combined with MIC and DTS, the proposed method achieves only a marginal improvement (an increase of 0.3 in Table 3), indicating limited effectiveness in these scenarios.

---

> ### Author Rebuttal · Authors · 2025-07-29
>
> Thank you for taking the time to share your comments and suggestions in the review assessment. We sincerely appreciate your positive feedback on our **rational and innovative approach**, **plug-and-play design**, **extensive experiments**, and **comprehensive ablation analysis**.
>
> We provide a detailed point-by-point response to your comments.
>
>
> ----
> **Q1:**  Increased training time due to the diffusion model
>
> **A1:**
>
>
> We fully acknowledge that incorporating a diffusion-based mechanism introduces additional training cost. However, we argue that this is a **reasonable and justified trade-off** for the following reasons:
>
> - As detailed in Appendix E (Table 6), the training time increases from 14.5h to 18.5h on DAFormer and from 28.9h to 34.8h on MIC. This represents a moderate increase (~27%), especially considering the consistent performance gains of up to +3.7 mIoU across multiple settings.
>
> - To reduce training cost, we propose a memory-efficient variant, DiDA($g_{time}$), which removes the diffusion encoder while preserving most of the performance (only 0.4 mIoU lower than full DiDA; see Table 2). This version reduces both memory consumption and training time significantly.
>
> - Importantly, DiDA incurs no additional inference cost, as the diffusion-related modules are used only during training. This ensures deployment efficiency is unaffected.
>
> - To further demonstrate DiDA’s efficiency, we compare its training cost and performance with other plug-in methods (e.g., FST, DTS) using the same DAFormer baseline and training schedule. Please refer to **A4**. Compared to FST and DTS, DiDA achieves better performance with lower training time and memory requirements, further validating its practicality.
>
> Overall, we believe the performance–efficiency trade-off is well-balanced, and DiDA remains scalable and practical for real-world applications.
>
> ----
>
> **Q2:**  Lack of comparison with recent state-of-the-art methods
>
> **A2:**
>
> We appreciate your suggestion. In Tables 3 and 4, we already compare DiDA against recent state-of-the-art UDA methods, including:
>
> - ADFormer and CoPT, both of which are ECCV 2024 methods.
> - Other recent methods such as MICDrop (also ECCV 2024) are also relevant, but our method **already surpasses** these works on the benchmarks we evaluate.
>
> Additionally, as DiDA is designed as a **plug-and-play module**, we focused on integrating it into widely-used baselines such as DAFormer, HRDA, and MIC. Notably, MIC remains one of the strongest open-source UDA methods available, and DiDA consistently improves its performance.
>
> We also note that DiDA, as a plug-in module, can be easily integrated into any future methods, which we plan to explore in follow-up work.
> In the final version, we will include comparisons with more recent methods, if they are released and relevant within the NeurIPS submission timeline.
>
>
> ----
>
>
> **Q3:**   Limited improvement when combined with  DTS
>
> **A3:**
>
> Thank you for raising this point. There appears to be a misunderstanding here.
>
> We would like to clarify that in Tables 1, 3, and 4, DiDA is **built directly on MIC**, not DTS. DTS is listed **for side-by-side comparison** as another plug-in method.
>
> Importantly, MIC is already a near-saturation UDA baseline, achieving 75.5 mIoU on GTA→CS. Our DiDA still manages to improve it to 76.8 mIoU (+1.3), which is non-trivial in this high-performance regime. As shown in prior works, gains >1 mIoU in such settings are considered meaningful and hard to achieve.
>
> To avoid confusion, we will revise the description in the final version to make the experimental design and integration strategy clearer.
>
> ----
>
>  **Q4:** Training time comparison with other methods
>
> **A4:**
>
> As already shown in Appendix E (Table 6), we provide a detailed comparison of training time, GPU memory usage, and iteration cost.
> To offer further comparison, we provide a direct training cost comparison with other plug-in methods such as FST and DTS built on the same baseline (DAFormer) and training schedule (40K iterations):
>
> | Method            | GPU Memory (MB) | Time per Iteration (s) | Total Training Time (h) | mIoU (%) |
> | ----------------- | --------------- | ---------------------- | ----------------------- | -------- |
> | DAFormer          | 9,807           | 1.32                   | 14.5                    | 68.3     |
> | +FST*             | 16,271          | 1.96                   | 21.9                    | 69.3     |
> | +DTS*             | 23,711          | 2.85                   | 31.3                    | 69.7     |
> | +DiDA($g_{time}$) | 12,754          | 1.59                   | 17.8                    | 69.9     |
> | +DiDA             | 15,856          | 1.64                   | 18.5                    | 70.3 |
>
> This table highlights that DiDA not only achieves **the best performance**, but also maintains **competitive training efficiency** and **lower memory usage** than DTS and FST. Especially, DiDA($g_{time}$) offers a compelling balance between cost and accuracy, making it **practical even under resource constraints**.
>
>
> ----
>
> We hope our response can resolve your concern. Please do not hesitate to let us know if you have further questions :)

---

> ### Comment · Area_Chair_Xipm · 2025-08-05
>
> Dear Reviewer xKyR,
>
> After reviewing the author's responses and the other reviews, could you share your updated thoughts following the rebuttal?
>
> Best,
>
> AC

---

> ### Comment · Reviewer_xKyR · 2025-08-05
>
> I would like to thank the authors for carefully and thoroughly addressing my questions. This rebuttal has resolved most of my concerns. For these reasons, I would like to raise my evaluation score to 5. I wish you good luck.

---

> > ### Author Response · Authors · 2025-08-07
> >
> > Dear Reviewer xKyR,
> >
> > Thank you so much for your encouraging feedback and for letting us know that most of your concerns have been resolved. We're truly grateful for your thoughtful evaluation and sincerely appreciate your decision to raise the score.
> >
> > We wish you all the best, and thank you again for your time and support!
> >
> > Best regards,
> >
> > Authors

---

> ### Comment · Reviewer_xKyR · 2025-08-05
>
> Thank you for reminding me. After reviewing the other authors' responses I see that the author has provided very detailed answers and addressed most of the questions. Therefore I am raising my evaluation score to 5.

---

### Official Review · Reviewer_Wf8D · 2025-07-04

**Clarity:** 3
**Significance:** 3
**Originality:** 3
**Rating:** 4
**Confidence:** 4

**Summary:**

The paper presents the DiDA Framework, an unsupervised domain adaptation approach for semantic segmentation. The insight idea of this paper is from the diffusion model, where the authors gradually add noise into the source and target domains (degradation steps) in order to remove domain-invariant features as domain differences gradually diminish. After that, the Semantic Shift Compensation module is proposed to mitigate the semantic shift problem during intermediate domain reconstruction. Overall, the paper's idea sounds interesting.

**Questions:**

Please refer to my weakness section.

Eqn. (7) has an extra parenthesis.

**Ethical Concerns:**

["NO or VERY MINOR ethics concerns only"]

**Final Justification:**

The rebuttal has addressed most of my concerns. Although the presentation of some parts in the paper could be further improved, the paper's idea sounds interesting, and the experimental results have illustrated the effectiveness of the proposed approach. Therefore, I would like to give a borderline accept.

**Limitations:**

Yes. The authors discuss the limitations in the appendix.

**Paper Formatting Concerns:**

There is no concern about the paper formatting. \

**Quality:**

3

**Strengths And Weaknesses:**

Strengths:
- The paper is well-motivated with interesting theoretical insight.
- The paper presentation is good.
- The idea of the proposed method is easy to catch via Figures 1 and 2.
- The experiments are solid and comprehensive.

Weaknesses
The method introduces four loss functions and a quite heavy end-to-end framework, which raises concerns about the stability of the framework.
- The demonstration of the intermediate domain is still vague, and hard to understand the flow of that module. It would be better it the authors further clarify this one.
- The motivation of Semantic Shift Compensation should be further explained.
- Did the authors discuss the trade-off between using larger $t$ and semantic information?. It means a larger timestep $t$ would reduce the domain gap; however, the semantic information is harder to learn. How to balance two factors? In addition, the timestamp is large, and the image becomes totally noisy. How can the model learn semantic segmentation from a totally noisy image?
- The inference step is not clear. How are compensatory features used in the inference step? According to Figure 2, it is fed directly into the segmentation head. Is it concatenated with any features or so?
- The authors mentioned ACDC in the experimental section, but I do not see the experiments of ACDC in the main paper. Please carry the experimental results of this benchmark into the main paper.

---

> ### Author Rebuttal · Authors · 2025-07-29
>
> Thank you for taking the time to provide detailed and constructive comments on our work. We greatly appreciate your positive feedback regarding the **interesting and intuitive idea**, **well-motivated theoretical insights**, **good presentation**, and **solid and comprehensive experiments**.
>
> We provide a detailed point-by-point response to your comments.
>
> ------
>
> **Q1:** Stability of The Framework
>
> **A1:**
>
> Thank you for highlighting this important concern. While DiDA introduces additional components to enhance domain adaptation, it has been **carefully designed to ensure training stability, modularity, and practical efficiency**. We address the concern from four perspectives:
>
> 1. **Controlled Complexity of Loss Functions:**
>    DiDA introduces only two additional loss terms beyond standard self-training:
>    - Degraded Image Consistency Loss ($L^D$), which regularizes predictions under degradation.
>    - Reconstruction Loss ($L^R$), which supervises semantic recovery.
>
>
>    Both are explicitly weighted in Eq. (10), and we conduct comprehensive ablation and sensitivity analysis in Appendix F     (Tables 7 and 8), showing that DiDA maintains stable performance across a wide range of hyperparameter values. These losses are complementary and do not introduce optimization instability.
>
>
> 2. **Modular and Loosely Coupled Components:**
>    As shown in Table 2, each module in DiDA can be independently removed or replaced without causing performance collapse. This modularity ensures robustness, and the architecture can be easily adapted or simplified depending on resource constraints.
>
> 3. **Efficiency and Lightweight Variant:**
>    We provide a memory-efficient variant, DiDA ($g\_{time}$), which removes the diffusion encoder and retains only the time embedding. Despite its simpler design, it achieves comparable improvements (see Appendix E, Table 6), reducing GPU memory usage by up to 20%.
>
> 4. **Inference-time Simplicity:**
>    Crucially, DiDA is used only during training. At inference, all auxiliary modules are disabled, and the model reverts to a standard segmentation network. This design ensures no additional latency, memory, or model complexity during deployment.
>
> 5. **Empirical Generalization Across Setups:**
>    DiDA demonstrates consistent and significant improvements across diverse architectures (CNNs and Transformers), datasets (GTA→CS, SYN→CS, CS→ACDC), and UDA baselines (DAFormer, HRDA, MIC), as shown in Tables 1, 3, and 4. This cross-domain robustness supports the stability of our design.
>
> While DiDA introduces two additional loss terms and auxiliary modules during training, the framework remains stable, modular, and efficient. The consistent performance gains across benchmarks validate its practical reliability.
>
> ------
>
> **Q2:** Clarification of Intermediate Domain Construction
>
> **A2:**
>
> Thank you for pointing out the need for further clarification regarding intermediate domain construction.
>
> The Degradation-based Intermediate Domain Construction module is based on the insight that **gradually degraded images create a continuum of distributions between source and target domains**. This strategy exploits the progressive removal of domain-specific features (e.g., texture) while retaining domain-invariant semantics (e.g., shape), thereby narrowing the domain gap in a controllable manner.
>
> To clarify the mechanism:
>
> - Figure 1  conceptually illustrates the forward diffusion process, where domain-specific information fades earlier (e.g., texture), and semantic shape information remains longer, under the guarantee of the theoretical insight (Section 3.2).
> - Section 3.3 formally defines these intermediate domains as samples from the forward process $x_t = \sqrt{\bar{\alpha}_t} x_0 + \sqrt{1 - \bar{\alpha}_t} \epsilon$, where $t$ controls degradation level.
> - Figure 4 shows performance trends across different degradation levels, empirically validating the effectiveness of intermediate domains.
> - Appendix H (Figure 7) presents a quantitative analysis: as degradation increases, the relative adaptation performance improves, suggesting that intermediate domains provide better alignment priors.
> - Appendix L shows image-level reconstructions across noise steps (Figures 9–12), visually confirming that core semantics are preserved even under significant degradation.
>
> We will revise the main paper to better explain the flow of degraded domain construction and directly reference these supporting sections.
>
> ------
>
> **Q3:** Motivation for Semantic Shift Compensation
>
> **A3:**
>
>
> Thank you for raising this fundamental question. The Semantic Shift Compensation module addresses a critical limitation in degradation-driven adaptation: **While degradation helps eliminate domain-specific cues, it may also corrupt domain-invariant semantics—especially under strong noise—leading to *semantic shift* and degraded predictions.**
>
> To mitigate this, we introduce a diffusion encoder ($g'$) and reconstruction head ($h'$), both conditioned on time-step embeddings. Together, they:
> - Perceive the extent of degradation (via time embedding),
> - Model the semantic shift caused by degradation,
> - Recover lost semantic structure (supervised by reconstruction loss $L^R$),
> - Regularize prediction consistency under noise (via $L^D$).
>
> This design allows the model to preserve meaningful features even under strong degradation, as shown by:
>
> - Appendix K (Figures 9–12): Visualizations of reconstructions under different degradation levels.
> - Appendix M: Comparison with existing domain bridging methods shows our approach performs better in preserving semantics while reducing domain gap.
> - Table 2 (Ablation): Removing this module leads to a noticeable drop in performance.
>
> We will enhance the main text to better illustrate this motivation and its benefits.
>
> ------
>
> **Q4:** Trade-off between Larger Timestep and Semantic Information
>
> **A4:**
>
> This is an excellent observation. Indeed, there is an intrinsic trade-off:
>
> - Small $t$ values: low degradation → retain semantics but maintain domain gap.
> - Large $t$ values: high degradation → reduce domain gap but risk semantic loss.
>
> **Rather than manually tuning this trade-off**, DiDA leverages random sampling of $t$ during training (uniformly from $1$ to $T$), similar to standard diffusion training. More importantly:
>
> - The time embedding module enables the model to perceive degradation level.
> - The diffusion encoder ($g'$) adapts feature representations accordingly.
> - The combined effect allows the model to implicitly learn how to balance domain invariance and semantic fidelity.
>
> Empirical evidence:
>
> - Figure 4(b) shows performance under fixed $t$: when $t$ is too large, semantic loss dominates, but the model performs best when $t$ matches degradation.
> - Appendix H (Figure 7) shows relative performance gradually improves, which means that the model’s adaptability is strengthened across different $t$.
> - Appendix K (Figure 10) confirms that the reconstruction is optimal when $t$ aligns with degradation.
>
> This design ensures that the model learns to adapt across a spectrum of domain proximity and semantic fidelity, without requiring manual balancing.
>
> ------
>
>  **Q5:** Inference Process and Use of Compensatory Features
>
> **A5:**
>
> Thank you for pointing this out. We are happy to clarify the inference procedure and the usage of the diffusion encoder ($g'$) and reconstruction head ($h'$).
>
> - **Inference:**
>   - During inference, only the backbone segmentation network $f_\theta = h \circ g$ is used.
>   - The diffusion-specific components—$g'$ and $h'$—are entirely removed at inference time.
>   - This means no additional computation, parameters, or architecture changes are introduced in deployment, ensuring full compatibility with existing segmentation pipelines.
>
>
> - **Training:**
>   - The diffusion encoder $g'$ is used to model the semantic shift introduced by degradation by conditioning on the time step $t$. Its output is **added** to the main encoder’s output ($g$) at multiple feature levels via residual connections.
>   - This residual fusion strategy avoids architectural disruption, helps preserve the original semantics, and makes the integration of $g'$ stable and lightweight.
>   - This fused feature is then passed to:
>     - the segmentation head $h$ for calculating the Degraded Image Consistency loss ($L^D$).
>     - the reconstruction head $h'$ for the Reconstruction loss ($L^R$).
>
>    This design ensures that:
>   - The original encoder $g$ remains intact and continues to learn normally.
>   - The auxiliary encoder $g'$ enhances the representation by compensating for semantic degradation in a plug-and-play fashion.
>   - The training remains stable due to learnable residual modulation rather than full replacement or concatenation, which could otherwise destabilize feature distribution.
>
>
> We will revise the main paper and Figure 2 to explicitly highlight this residual fusion mechanism and clarify that no concatenation or architectural modification occurs during inference.
>
> ------
>
> **Q6:** Missing ACDC Results
>
> **A6:**
>
> Thank you for noticing this. The ACDC benchmark is indeed important as it evaluates domain shifts due to adverse weather.
> - The main results for CS→ACDC are summarized in Table 1 (mIoU scores).
> - Detailed per-class results are included in Appendix D, Table 5.
>
> We agree that this benchmark deserves greater visibility. In the final version, we will move the key ACDC results and analysis into Section 4, ensuring clarity and completeness.
>
> ------
>
> **Q7:** Typo
>
> **A7:**
>
> Thank you for pointing out this formatting issue. We will correct the extra parenthesis in Equation (7) in the final version.
>
> ------
>
> We hope our response can resolve your concern. Please do not hesitate to let us know if you have further questions :)

---

> > ### Comment · Reviewer_Wf8D · 2025-08-04
> >
> > I would like to thank the authors for their rebuttal. The rebuttal has addressed most of my concerns. I suggest the authors incorporate the answers of **A5** into the paper and also bring the experimental results of ACDC into the main paper. In addition, the presentation of the motivation for Semantic Shift Compensation should be better explained in the revised version. Based on these reasons, I would like to raise my rating to borderline accept.

---

> > > ### Author Response · Authors · 2025-08-05
> > >
> > > Dear Reviewer Wf8D,
> > >
> > > Thank you so much for your encouraging feedback and helpful suggestions! We're glad to hear that most of your concerns have been addressed.
> > >
> > > We'll make sure to include the answers from A5 in the paper, bring the ACDC results into the main content, and improve the explanation of the motivation behind Semantic Shift Compensation.
> > >
> > > We really appreciate your time and support.
> > >
> > > Best regards,
> > >
> > > Authors

---

### Note · Authors · 2025-08-14

We sincerely thank the AC and reviewers for their constructive feedback and valuable discussions. We are encouraged by the positive recognition of our work’s **motivation, theoretical grounding, and consistent empirical performance across diverse UDA settings.**

------

**Key Contributions**

Our work introduces **DiDA**, a novel domain bridging framework for UDA in semantic segmentation:

1. **Degradation-based Intermediate Domain Construction**: We propose a principled approach to reduce domain discrepancy via progressive image degradation, forming a continuous bridge between source and target domains in both image and feature space.
2. **Semantic Shift Compensation**: A time-conditioned diffusion encoder is introduced to recover semantic information lost during degradation, preserving discriminative representations in intermediate domains.
3. **Modular Design and Zero-Cost Inference**: DiDA is a plug-and-play framework that integrates with existing UDA methods and introduces no extra inference-time overhead.

------

**Concerns Addressed**

1. **Framework Stability**: We clarified DiDA’s design via ablations and sensitivity analysis, and introduced a lightweight variant with comparable performance and reduced training cost.
2. **Theoretical Motivation**: We grounded our approach in the attribute-loss dynamics of diffusion processes (inspired by [75]), and extended this insight to domain adaptation. We clarified this as a task-specific reinterpretation, not a new theorem.
3. **Intermediate Domain Visualization**: We provided empirical evidence (Appendix H) and t-SNE plots (to be included in final version) showing increased feature alignment between domains under degradation.
4. **Inference Mechanism**: We clarified that compensatory features are fused during training via residual connections, with diffusion modules removed at inference.
5. **Cross-City Evaluation**: Following reviewer suggestion, we added results on Cityscapes→Oxford RobotCar, showing consistent gains (+1.7~1.8 mIoU) on both classic and modern backbones.
6. **Clarity and Terminology**: We will revise terminology and figures (e.g., Fig. 2) to improve precision.

------

We believe DiDA provides a novel, theoretically grounded, and practically effective solution to bridging domain gaps via image degradation, and we hope it contributes meaningfully to advancing research in UDA.

Thank you again for your time, attention, and thoughtful evaluation throughout the review process.

---

### Decision · Program_Chairs · 2025-09-17

**Decision:**

Accept (poster)

**Comment:**

This paper presents DiDA, an unsupervised domain adaptation (UDA) framework for semantic segmentation that leverages image degradation to reduce domain gaps. The method creates intermediate domains through progressive degradation (e.g., noise, blur) and uses a diffusion-based encoder to preserve discriminative features while compensating for semantic shifts.

Strengths

* Demonstrates consistent improvements over state-of-the-art UDA baselines across GTA→Cityscapes, SYNTHIA→Cityscapes, and cross-city benchmarks (e.g., Cityscapes→Oxford RobotCar).
* Integrates smoothly with existing approaches (e.g., DAFormer, HRDA) without inference overhead.
* Offers an innovative reinterpretation of diffusion dynamics for domain alignment, supported by thorough ablations and new cross-city evaluations.

Weaknesses & Resolutions

* *Theoretical originality*: Initially questioned, but partially addressed by clarifying the adaptation of diffusion theory to UDA. Core contribution remains application-driven.
* *Empirical validation*: Strengthened through additional experiments, t-SNE results confirmed improved feature alignment, and a lightweight variant reduced memory usage by 20% with negligible performance loss.
* *Clarity & stability concerns*: Largely resolved in the rebuttal.


While some lingering concerns remain (e.g., theoretical nuances noted by Reviewer 1mYM), reviewers generally acknowledged the framework’s robust empirical results and practical contribution. Authors are encouraged to incorporate the rebuttal into the final revision and address the lingering concerns.


DiDA provides a practical, theoretically informed approach to domain adaptation, validated across diverse settings. Although its theoretical foundation builds on existing diffusion principles, the novel UDA-specific application and consistent performance gains strongly support its acceptance.